# Electrocorticography is superior to subthalamic local field potentials for movement decoding in Parkinson's disease

Timon Merk[1]*, Victoria Peterson[2,3], Witold J Lipski[4], Benjamin Blankertz[5], Robert S Turner[4], Ningfei Li[1], Andreas Horn[1], Robert Mark Richardson[2,3†], Wolf-Julian Neumann[1]*†

[1]Movement Disorder and Neuromodulation Unit, Department of Neurology, Charité - Universitätsmedizin Berlin, corporate member of Freie Universität Berlin and Humboldt Universität zu Berlin, Berlin, Germany; [2]Brain Modulation Lab, Department of Neurosurgery, Massachusetts General Hospital, Boston, United States; [3]Harvard Medical School, Boston, United States; [4]Department of Neurobiology, University of Pittsburgh, Pittsburgh, United States; [5]Department of Computer Science, Technische Universität Berln, Berlin, Germany

*For correspondence:
timon.merk@charite.de (TM);
julian.neumann@charite.de (W-JN)

†These authors contributed equally to this work

Competing interest: The authors declare that no competing interests exist.

**Abstract** Brain signal decoding promises significant advances in the development of clinical brain computer interfaces (BCI). In Parkinson's disease (PD), first bidirectional BCI implants for adaptive deep brain stimulation (DBS) are now available. Brain signal decoding can extend the clinical utility of adaptive DBS but the impact of neural source, computational methods and PD pathophysiology on decoding performance are unknown. This represents an unmet need for the development of future neurotechnology. To address this, we developed an invasive brain-signal decoding approach based on intraoperative sensorimotor electrocorticography (ECoG) and subthalamic LFP to predict grip-force, a representative movement decoding application, in 11 PD patients undergoing DBS. We demonstrate that ECoG is superior to subthalamic LFP for accurate grip-force decoding. Gradient boosted decision trees (XGBOOST) outperformed other model architectures. ECoG based decoding performance negatively correlated with motor impairment, which could be attributed to subthalamic beta bursts in the motor preparation and movement period. This highlights the impact of PD pathophysiology on the neural capacity to encode movement vigor. Finally, we developed a connectomic analysis that could predict grip-force decoding performance of individual ECoG channels across patients by using their connectomic fingerprints. Our study provides a neurophysiological and computational framework for invasive brain signal decoding to aid the development of an individualized precision-medicine approach to intelligent adaptive DBS.

## Editor's evaluation

This paper evaluates movement decoding from intracranial brain recordings in patients with Parkinson's disease. Interestingly, the authors demonstrate that cortical recordings (electrocorticography) outperform subthalamic nucleus in decoding grip force. This work will be of interest to brain computer interface, movement disorder, motor control, and general neurophysiology communities.

## Introduction

Subthalamic deep brain stimulation (DBS) for Parkinson's disease (PD) is one of the most successful neurotechnological advances in translational neuroscience to date. In addition to its clinical utility, DBS has provided unique insight into the neurophysiology of movement disorders (*Cagnan et al., 2019*; *Krauss et al., 2021*). PD has been associated with increased beta synchronization and beta bursts in the basal ganglia (*Kühn et al., 2006*; *Neumann et al., 2016*; *Kehnemouyi et al., 2021*) and exaggerated phase amplitude coupling and waveform sharpness asymmetry in cortex (*de Hemptinne et al., 2015*; *Cole et al., 2017*). Symptom severity in the OFF medication state was shown to correlate with resting beta power in the STN across patients (*Kühn et al., 2006*; *Neumann et al., 2016*). Such observations have inspired the idea of adaptive DBS (aDBS), where electrophysiological signals are used to change stimulation parameters in response to evolving clinical states (*Little et al., 2013*; *Beudel and Brown, 2016*; *Tinkhauser et al., 2017*; *Swann et al., 2018*; *Piña-Fuentes and van Dijk, 2019*; *Velisar et al., 2019*; *Hwang et al., 2020*; *Petrucci et al., 2020*). In a series of seminal papers it was shown that significant clinical benefit and reduced side-effects could be achieved, when stimulation was triggered by beta power (*Little et al., 2013*; *Velisar et al., 2019*). Machine-learning for aDBS applications can integrate multivariate feature sets for adaptive DBS control beyond beta power. First trials on machine learning based movement classification to trigger adaptive DBS either using electrocorticography (ECoG) or subcortical local field potentials (LFP) in essential tremor have shown promising results (*Opri et al., 2020*; *He et al., 2021*). In the future, smart implants may become available that combine invasive brain signal decoding with real-time stimulation adaptation, toward a precision medicine approach to adaptive DBS in PD and other brain disorders. However, the identification of optimal decoding strategies and the characterization of relevant factors with impact on decoding performance remains and unmet need. With the present study, we address this by a thorough investigation grip-force decoding that is motivated by the well-described relationship of vigor, movement velocity, bradykinesia, and dopamine in Parkinson's disease (*Turner and Desmurget, 2010*; *Yttri and Dudman, 2016*; *Lofredi et al., 2018*). We use state-of-art machine learning algorithms with multimodal invasive neurophysiology and whole-brain connectomics in PD patients undergoing DBS electrode implantation. Our results highlight the utility of cortical vs. subcortical signals to accurately decode grip-force and establish a link between decoding performance and motor impairment in PD. Finally, we investigate brain networks from ECoG recording locations with normative structural and functional connectomics and demonstrate the predictive power of connectomic fingerprints for brain signal decoding.

## Results

### Real-time processing and feature definition

We analyzed sensorimotor ECoG and subthalamic LFP data recorded intraoperatively from 11 PD patients undergoing DBS implantation during performance of a Go/No-Go based cued grip-force task (*Figure 1A*). Individual electrode localizations in Montreal Neurological institute (MNI) space are shown in *Figure 1B* with typical responses (*Kühn et al., 2004*; *Androulidakis et al., 2007*; *Kondylis et al., 2016*; *Lofredi et al., 2018*) in *Figure 1C* aligned to onset of grip force (total n=2685, on average n=244 ± 149 STD movements per patient, see *Figure 1—figure supplement 1* for more detail on grip-force variability). For the use in machine-learning models, band power feature time-series were extracted in a real-time BCI compatible implementation (*Figure 1D*) streamed in virtual packets of 100ms length at a sampling rate of 1000 Hz to mimic the online application. Variance as a measure of amplitude of rereferenced, band-pass filtered raw data segments was extracted at 10 Hz with an adaptive window length from 1000 to 100ms of past data for eight oscillatory features [θ (4–8 Hz), α (8–12 Hz), β (13–35 Hz), low β (13–20 Hz), high β (20–35 Hz), low γ (60–80 Hz), high-frequency activity (HFA) (90–200 Hz) and all γ (60–200 Hz)]. All features were normalized to the median of the past 10 s to compensate for potential signal changes over time. The target variable was continuously measured grip-force (z-scored for each recording session), which was cleaned from noise and baseline drift (*Xie et al., 2018*).

### Including preceding signals up to 500 ms before the decoded sample improves grip-force decoding performance

A linear model analysis of all eight oscillatory features per channel was used to investigate the contributing band power correlations for time-points simultaneous to and preceding target samples

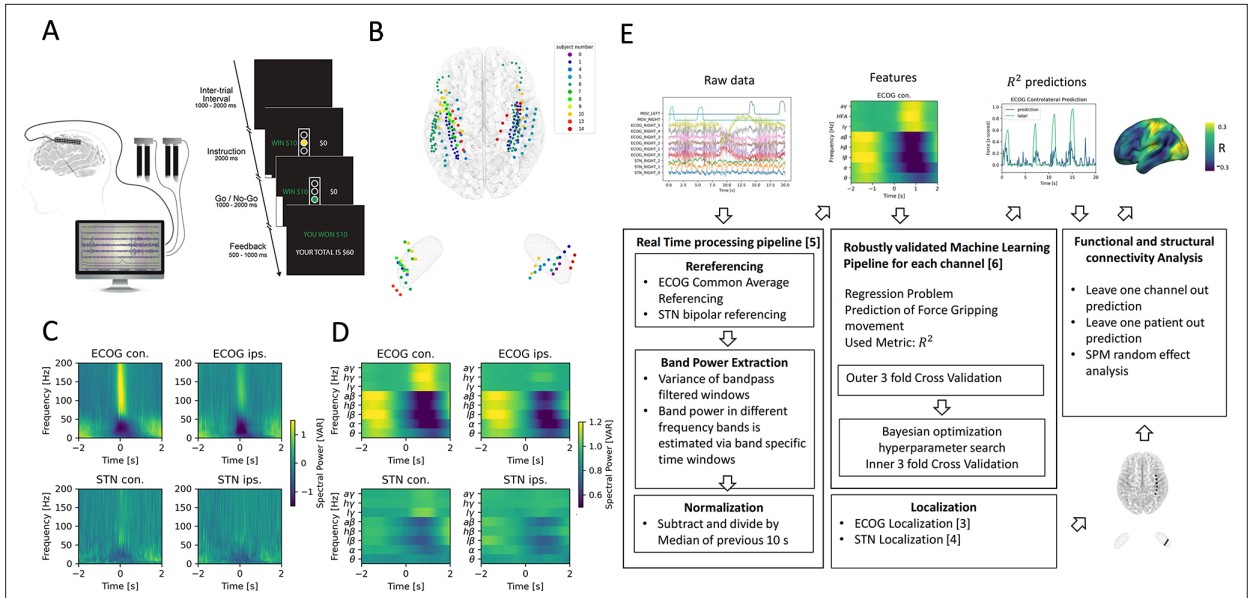

**Figure 1.** Movement induced spectral changes are more dominant for ECoG than STN-LFP signals for a grip force task before and after a machine learning feature signal processing pipeline. (**A**) ECoG, STN, and gripping force were recorded simultaneously during performance of a Go / No-Go task. (**B**) Individual ECoG and STN electrodes were localized and transformed into in Montreal Neurological Institute (MNI) space (*Figure 1—source data 1*). Note that ECoG strip designs varied slightly between patients (see *Supplementary file 1a*), leading to varying dimensions of overall input feature matrices. The number of ECoG channels (average n=9.45 ± 11.15 STD per hemisphere) is higher compared to the number of STN LFP channels (n=3). (**C**) Mean spectral power of all ECoG and STN channels for contra- and ipsilateral movements showed typical movement induced spectral changes (*Figure 1—source data 2*). (**D**) Virtual streaming of data packets secured real-time compatible processing and normalization to extract time-frequency modulations into discrete feature time-series. Mean features of all ECoG and STN channels are visualized (*Figure 1—source data 3*). (**E**) Schematic flow chart of the implemented real-time enabled feature extraction, machine learning evaluation and functional and structural connectivity analysis pipeline.

The online version of this article includes the following source data and figure supplement(s) for figure 1:

**Source data 1.** ECoG and STN electrode localizations.

**Source data 2.** Mean ECoG and STN spectral power.

**Source data 3.** Mean ECoG and STN features.

**Figure supplement 1.** Analyzed movements show variability in maximum amplitude and velocity.

of continuous grip-force measurements. *Figure 2A* shows the weight distributions of multivariable linear models of the best performing channels per subject. Since each cortical or STN electrode has multiple channels, only the best channel per electrode is selected for this visualization. As the interpretability of coefficients in multivariable models is limited (*Haufe et al., 2014*), we have further visualized the normalized coefficients of univariate models for each relative time-point and frequency band in *Figure 2B*. Next, to investigate the cumulative performance contribution of preceding time points for optimal feature construction, all frequency bands were concatenated while continuously increasing the cumulative number of premovement time-points (from –100 to –1000ms) and each set was subjected to training a Wiener Filter. The respective best channel $R^2$ performances are shown in *Figure 2C*. A performance saturation becomes visible when concatenating 5 time-points from 500ms (prior to target sample) to 0ms (target sample), resulting in an optimal input vector of 8 frequency bands with 5 time-points (=40 features) for further analyses.

## XGBOOST outperforms other machine learning models for grip-force decoding

In order to build a grip-force decoder, different machine learning (ML) algorithms were tested in a large-scale Bayesian Optimization hyperparameter search (see *Supplementary file 1B* for a list of hyperparameters for each model). Elastic - Net regularized Linear Models, Neural Networks and Gradient Boosted trees (XGBOOST) (*Chen and Guestrin, 2016*) were tested for each channel for contra- and ipsilateral movements. XGBOOST was included as it can learn non-linearities and has

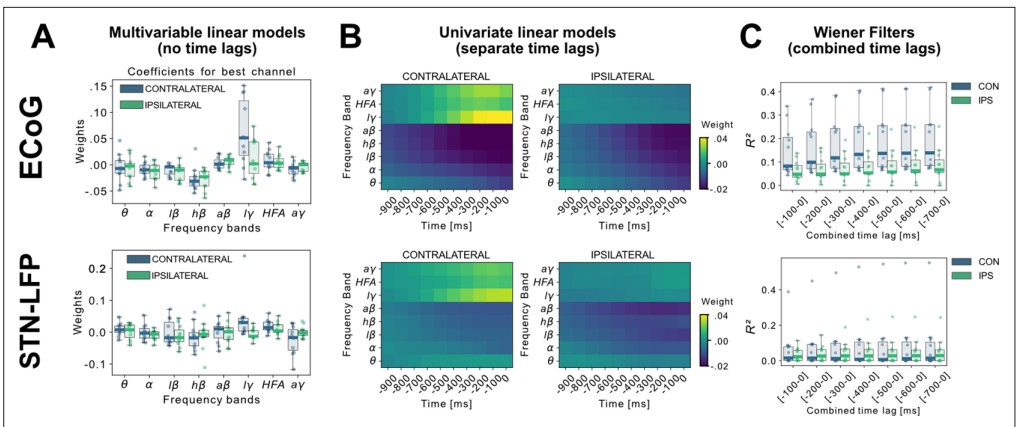

**Figure 2.** Linear Models and Wiener Filters reveal temporally and spectrally specific coefficient distributions with grip-force decoding performance gain by including signals preceding the target sample by up to 500ms. (**A**) Multivariable linear model coefficients trained only from the instantaneous sample (0 time lag with respect to decoded target sample) including all frequency bands from best channels per patient resemble movement induced spectral changes with beta desynchronization and gamma synchronization (*Figure 2—source data 1*). ECoG derived coefficients yield higher absolute values than STN-LFP derived coefficients. (**B**) Univariate frequency and time lag specific Linear Models were trained and visualized to improve interpretability of average coefficients in the absence of interactions (*Figure 2—source data 2*). Low γ (60–80 Hz), HFA (90–200 Hz), and all γ (60–200 Hz) bands show stronger positive associations for contralateral over ipsilateral movements. Moreover, stronger associations are visible for ECoG over STN-LFP signals for $\beta$, HFA, and $\gamma$ bands. (**C**) Wiener Filters can integrate multiple time-steps in Linear Models leading to an incremental performance gain when signals are included preceding the current target sample by up to 500ms (*Figure 2—source data 3*).

The online version of this article includes the following source data for figure 2:

**Source data 1.** Best channel Linear Model coefficients trained from instantaneous sample.

**Source data 2.** Univariate Linear Model coefficients of single frequency band and time lag.

**Source data 3.** Wiener Filter multiple time-step comparison.

advantages over other models with respect to feature selection. To further utilize potential information derived from spatial patterns, the Source Power Comodulation (SPoC) framework (*Dähne et al., 2014*) was used in combination with Elastic - Net or XGBOOST predictors. Each model was informed by 40 features (8 specific frequency bands concatenated at 5 time-points ranging from t = –500ms to t=0ms to the target sample) per channel and evaluated via rigorously cross-validated test-set predictions ranked by $R^2$ coefficients of determination. *Figure 3* shows performance outcomes for the different machine learning methods, with overall best results achieved by XGBOOST from ECoG signals (see *Supplementary file 1c* for further details). Contralateral ECoG strips had significantly higher decoding performances than ipsilateral ones (contralateral $R^2 = 0.31 \pm 24$, ipsilateral $R^2 = 0.13 \pm 0.16$, p=0.02). Given the relatively low decoding performances for STN-LFP, we applied permutation tests to confirm that performance was above chance (contralateral p=0.025, ipsilateral p=0.028). Corroborating the model choice in previous literature, highest STN performances were achieved with the Wiener Filter method for contra- and ipsilateral movements (*Shah et al., 2018*). Importantly, varying combinations of multiple ECoG and/or STN channels did not lead to significant performance advantages (*Figure 3* C+D), which is important for the utility and design of machine learning enabled implantables.

## Grip-force decoding performance is correlated with PD motor impairment and subthalamic beta burst dynamics

To investigate potential sources of bias from patient specific information on grip-force decoding performance, we performed Spearman's correlations with the grand average from all contra -and ipsilateral decoding performances. Averaging was necessary to obtain one value per patient. Age ($\rho$ =–0.16, p=0.32), disease duration in years ($\rho$ =0.31, p=0.17) and number of movements ($\rho$ =–0.41, p=0.11) and movement variability (Rho = –0.49, p=0.06) did not reveal significant correlations.

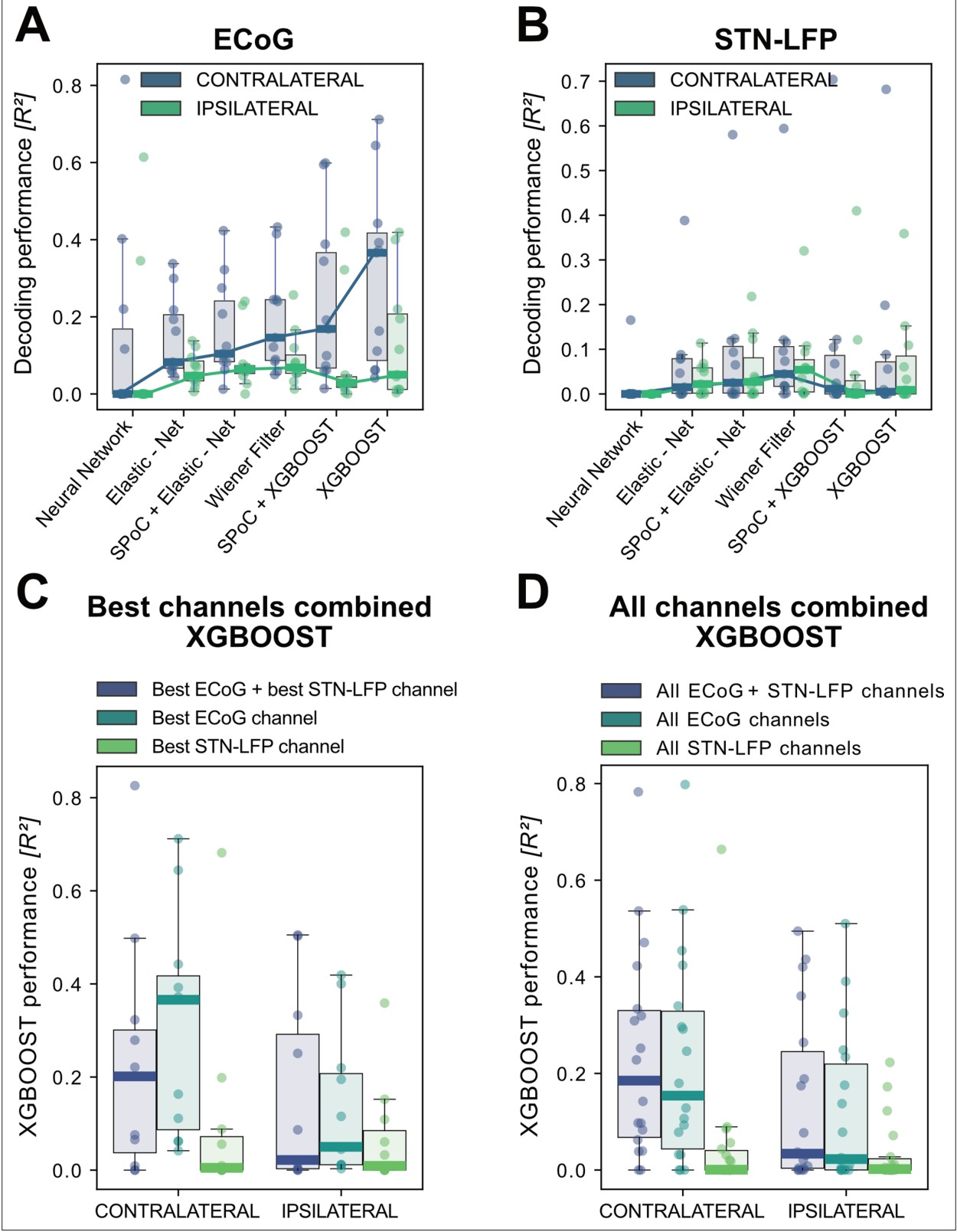

**Figure 3.** XGBOOST outperforms other machine learning methods for ECoG based grip-force decoding. Based on the presented real-time compatible signal processing pipeline Neural Networks, Elastic - Net regularized Linear Models, Wiener Filters and extreme Gradient Boosting (XGBOOST) regression models were tested. Mean $R^2$ test-set grip-force decoding performances are shown for the best channel per patient after 10 rounds of Bayesian Optimization of hyperparameters with nested cross-validation for ECoG (**A**) and STN-LFP (**B**) (*Figure 3—source data 1*). The same pipeline

*Figure 3 continued on next page*

*Figure 3 continued*

was subjected to spatial feature extraction approach using all available channels of an electrode for each patient with Source Power Comodulation (SPoC). Best ECoG (**A**) performances were obtained by XGBOOST regressors. STN-LFP signals (**B**) did not exhibit performance gain when applying advanced machine learning methods. The mean ECoG vs. STN XGBOOST performance differences of contralateral $\Delta R^2$ = 0.21 ± 0.18 and ipsilateral $\Delta R^2$ = 0.069 ± 0.08 movements, indicate the higher grip-force decoding performance of ECoG over STN signals. The mean test-set prediction performances were higher for ECoG than for STN-LFP signals across all patients, for both contra- and ipsilateral movements. Best ECoG channels outperformed best STN-LFP channels and the combination of best channels from both ECoG and STN-LFP (**C**) (*Figure 3—source data 2*). When combining multiple channels, performances improve through the combination of ECoG and STN-LFPs (**D**), but the performances remain below individual best ECoG channels as depicted in (**C**). For combined ECoG +STN – LFP training, the model learned specific combinations between both feature locations and failed to select only the best ECoG features due to overfitting.

The online version of this article includes the following source data for figure 3:

**Source data 1.** Cross-validated ECoG and STN machine learning model performances for single channels.

**Source data 2.** Cross-validated combined and best-channel XGBOOST performances for best ECoG and STN channels.

**Source data 3.** Cross-validated XGBOOST performances for multichannel models based on ECoG, LFP and combined ECoG-LFP channels.

---

We further investigated whether motor impairment related to the hypodopaminergic state in PD can explain differences in grip-force decoding across patients. Therefore, we correlated preoperative OFF medication total UPDRS-III scores, which revealed negative correlations for best ECoG ($\rho$ =–0.55, p=0.039) and STN-LFP ($\rho$ =–0.55, p=0.042) channels (*Figure 4* A+B). Combined ECoG and STN channel performance also showed significant correlations ($\rho$=–0.54, p=0.045), as well as combined ECoG ($\rho$=–0.55, p=0.045) and combined STN-LFP performances ($\rho$=–0.61, p=0.024). To test whether the correlation measure was corrupted by outliers, we repeated the analysis using the robust percentage-bend correlation (*Pernet et al., 2012*) which replicated the significant association between UPDRS total score and mean contra -and ipsilateral channel performance for ECoG ($r$=–0.62, p=0.04) and STN ($r$=–0.7, p=0.016). This correlation was temporally specific to decoding of ongoing grip-force, indicative of the models' underestimation of motor output (*Figure 4C*). Thus, the lower decoding performance in patients with more severe symptom severity could not be attributed to changes in decoder output in the absence of movement or temporal imprecision. This has practical implications and highlights the importance of investigating interactions between disease and machine learning approach for neural implants.

To better understand the relationship of PD pathophysiology and grip-force decoding performance, we have further investigated associations between cortical and subthalamic beta burst dynamics. We follow the methodology of previous reports that demonstrated that the time spent in beta burst correlates with impairment of movement kinematics (*Torrecillos et al., 2018*). Beta bursts were defined as threshold crossings of the beta feature vector above the 75th percentile of the baseline period. Following the previous finding that specifically the time-spent in low-beta but not high-beta bursts was correlated with PD motor impairment (*Lofredi et al., 2019*), we investigated these bands separately for the motor preparation period (−1–0 s with respect to movement onset) and movement execution period (0–1 s following movement onset). To uncover a potential relationship of the beta-burst metric with PD pathophysiology, we performed correlations with UPDRS-III total scores. Significant correlations were found between UPDRS-III and low-beta bursts in STN-LFP signals during motor preparation ($\rho$=0.63, p=0.02; *Figure 5A*) and movement execution ($\rho$=0.56, p=0.04; data not shown), but not for the high-beta band (p>0.05). Conversely, for ECoG high-beta but not low-beta burst dynamics during motor preparation but not movement periods were significantly correlated with UPDRS-III total scores ($\rho$=0.55, p=0.04). In summary, we provide evidence that both subthalamic and cortical beta burst dynamics relate to PD motor sign severity with subthalamic low-beta bursts showing the most robust correlations, both during motor preparation and movement periods.

To relate these findings to movement decoding performance from cortex, we correlated the grand average XGBOOST grip-force decoding performances from ECoG channels (as above for UPDRS-III) with high- and low-beta burst dynamics in both ECoG and STN-LFP signals. ECoG based grip-force decoding performance was significantly correlated with subthalamic low-beta burst dynamics during motor preparation ($\rho$=–0.76, p=0.004) and movement execution ($\rho$=–0.71, p=0.005; *Figure 5B*). Subthalamic burst dynamics in the high-beta band also correlated with ECoG decoding performances during movement ($\rho$=0.71, p=0.007) but not motor preparation. Cortical burst dynamics from ECoG

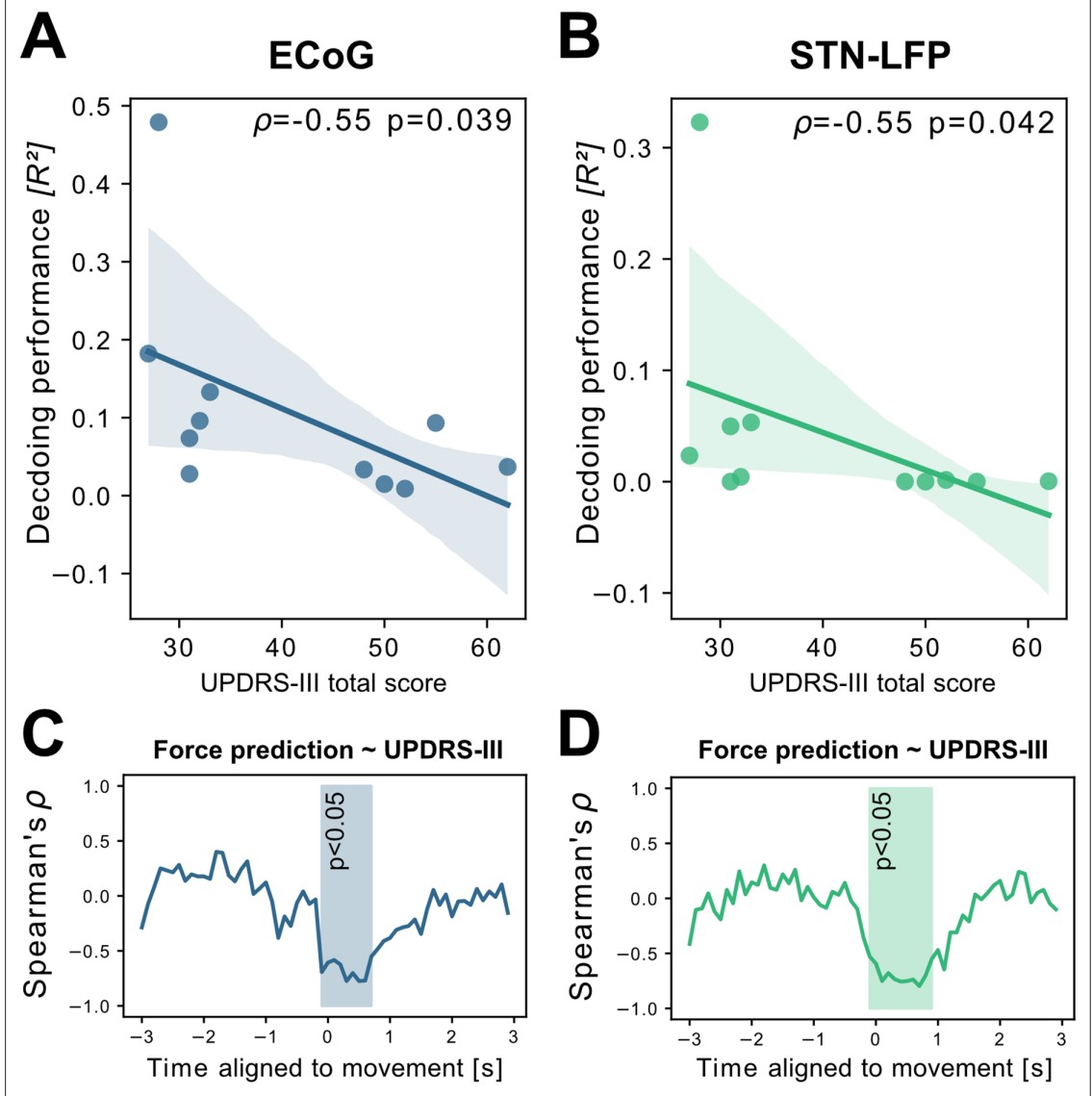

**Figure 4.** Grand average grip-force decoding performances correlate inversely with preoperative PD motor sign severity. UPDRS-III scores show significant negative correlations with patient-wise XGBOOST grip-force decoding performance averages for (**A**) ECoG ( $\rho$ =–0.55, p=0.039) and (**B**) STN-LFP signals ( $\rho$ =–0.55, p=0.042) (*Figure 4—source data 1*). The temporal specificity of this correlation is revealed through movement aligned sample-wise correlations of average force prediction model output with UPDRS-III scores across patients (cluster based corrected significant segments are displayed shaded) (C+D) (*Figure 4—source data 2*).

The online version of this article includes the following source data for figure 4:

**Source data 1.** ECoG and STN single channel performances and UPDRS ratings.

**Source data 2.** ECoG and STN Force prediction UPDRS correlation.

signals did not reveal significant correlations with ECoG-based grip-force decoding performances. Relevant correlations alongside exemplar burst visualizations and corresponding grip-force decoding traces are shown in *Figure 5*.

## Brain mapping of grip-force decoding performance from invasive cortical and subthalamic recordings

The spatial distributions of decoding performance on cortex and STN for contra- and ipsilateral movements are shown in *Figure 6*. To evaluate the relevance of recording location with respect to decoding performance, we calculated correlations of performance measures with a priori defined implantation

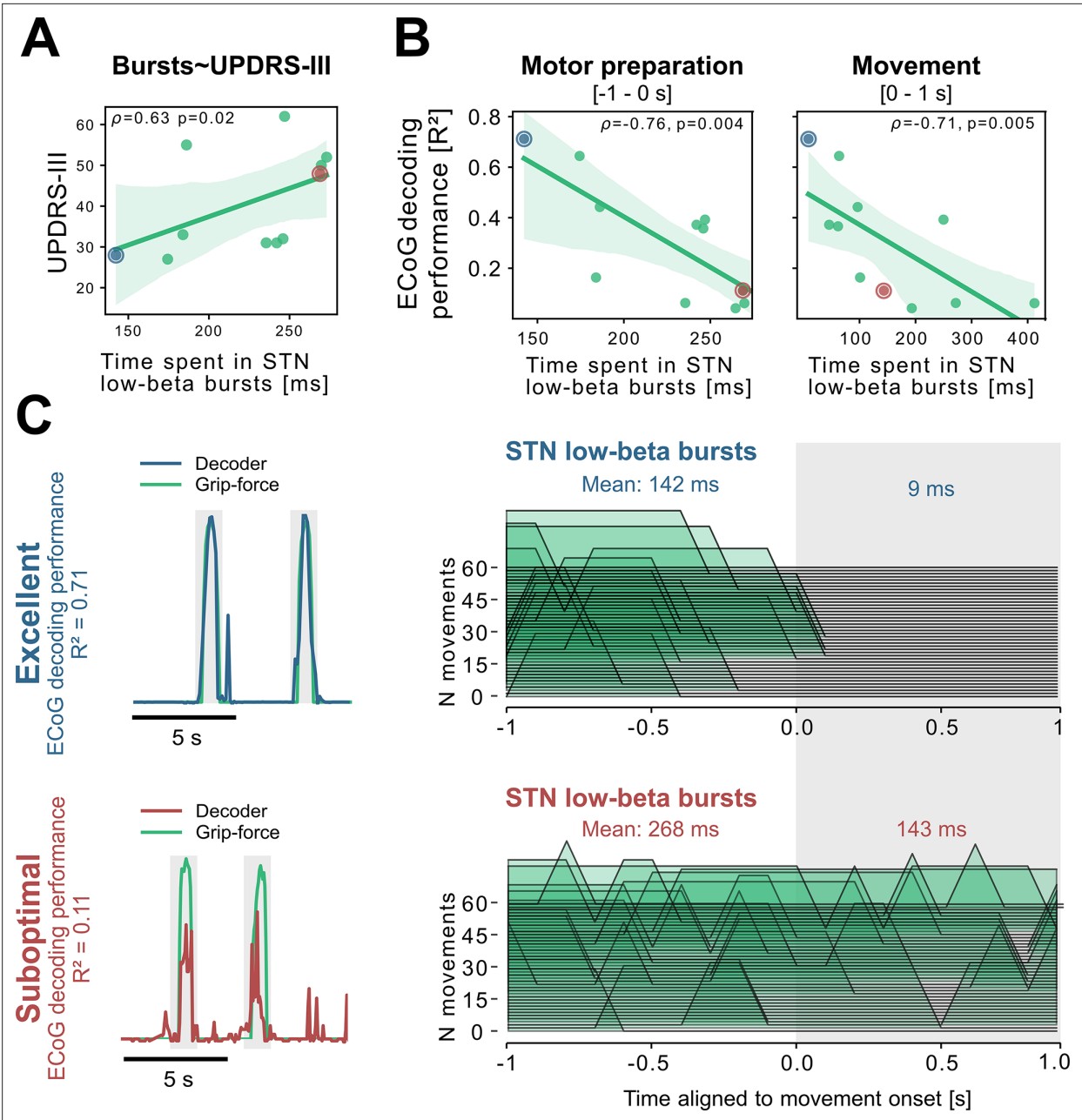

**Figure 5.** Subthalamic low-beta bursts relate to PD motor impairment and are associated with lower ECoG decoding performance. UPDRS-III scores are significantly correlated with time spent in subthalamic low-beta bursts in the motor preparation period (**A**) and during movement (not shown). Average XGBOOST decoding performance correlated inversely with time spent in subthalamic low-beta bursts during motor preparation and movement performance (**B**) (*Figure 5—source data 1*). Patient examples with excellent (R²=0.71; blue) and suboptimal (R²=0.11; red) performances are highlighted in (**B**) and shown in further detail in (**C**) (*Figure 5—source data 2*). Note the difference in decoder output with respect to the original grip-force trace (left panel) and the differences in burst frequencies and durations across movement repetitions (right panel) in the motor preparation and movement execution (grey shaded area) period.

The online version of this article includes the following source data for figure 5:

**Source data 1.** Time spend in low-beta burst performance and UPDRS correlation.

**Source data 2.** Movement onset aligned low-beta bursts for two subjects.

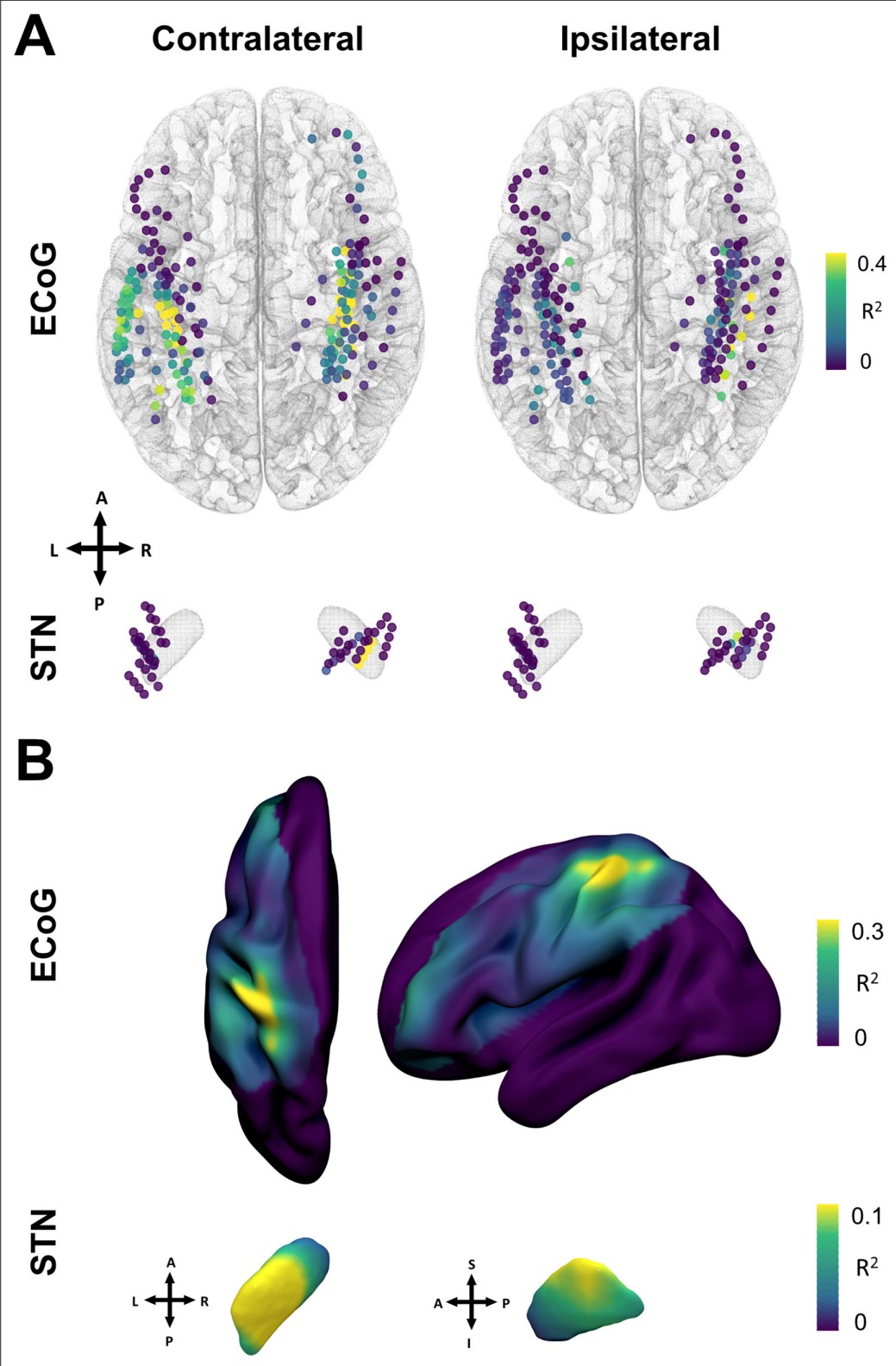

**Figure 6.** Grip-force decoding performances spatially peak in sensorimotor cortex and the dorsolateral STN. (**A**) Channels are color coded for individual XGBOOST grip-force regression performances per channel. Performance differences shown are in favor of ECoG over STN and contralateral over ipsilateral recording locations for movement decoding. (**Figure 6—source data 1**) (**B**) Spatial interpolation across all contacts projected to the

*Figure 6 continued on next page*

*Figure 6 continued*
left hemisphere shows peak performances in sensorimotor cortex. STN interpolated decoding performance peaks in the dorsolateral portion of the STN, in proximity to the best therapeutic target (*Caire et al., 2013*).

The online version of this article includes the following source data for figure 6:

**Source data 1.** Single channel XGBOOST coordinates and performances.

targets, namely the dorsolateral STN (*Caire et al., 2013*; *Horn et al., 2017a*) and the hand-knob of the precentral gyrus (*Mayka et al., 2006*). Linear mixed effects models showed a significant within-subject relation for contralateral ECoG decoding performances ($\beta$=−0.002, Lower CI = −0.003, upper CI = −0.001, $R^2$ = 0.57, p<0.001), but not STN locations (p>0.05). The dependent variable was the decoding performance, the fixed effect was the distance to hand knob area or dorsolateral STN respectively, and the random effect the subject. Repeating the analyses across electrodes and patients in a cross-validated manner revealed no significant predictive value (p>0.05). Thus, Euclidean distance to hand knob area for ECoG and therapeutic target for STN was significantly correlated with decoding performance within patients, but could not predict decoding performance across channels or patients.

## Whole-brain connectomics can aid the discovery of brain networks underlying the neural encoding of grip-force

The ability to account for decoding performances for invasive electrodes may soon become as important as accounting for variance in stimulation effects, as bidirectional clinical brain computer interfaces will rely both on electrical sensing and stimulation. Recently, network mapping of neurostimulation targets has shown utility to predict variance in clinical outcomes following DBS (*Horn et al., 2017b*; *Horn and Fox, 2020*; *Li et al., 2020*). Here, we extended the same framework to predict variance in grip-force decoding performance observed from single channels, using the XGBOOST grip-force decoding results. In this approach – termed *prediction network mapping* – we calculated functional and structural connectivity *fingerprints* by projecting each recording location to a group connectome that was acquired in a cohort of PD patients. These fingerprints denote to which other brain areas each site is connected to. Using a discriminative fiber tracking analysis, (*Baldermann et al., 2019*; *Li et al., 2020*) we analyzed the predictive value of structural connectivity from ECoG recording locations (for an exemplar case see *Figure 7A*) for XGBOOST decoding performance. Therefore, diffusion imaging derived whole-brain fiber connectome data traversing to more than 20% of recording locations were used (*Figure 7B*). The specific fiber distributions included structural projections spanning sensory, motor and prefrontal cortex, and could significantly predict decoding performance of left out channels ($\rho$ =0.38, p<0.0001; thresholded at a false discovery rate $\alpha$=0.05) and patients ($\rho$ =0.37, p<0.0001) in a cross validated manner (*Figure 7D*). Next, we created spatial models of optimal decoding performance for functional connectivity (R-Maps are shown in *Figure 7C*). This model led to significant predictions of decoding performance in leave-one-channel-out ($\rho$=0.37, p<0.0001) and leave-one-subject-out cross validations (functional connectivity $\rho$ = 0.37, p<0.0001) (*Figure 7E*). The results were further validated with voxel-wise correlations using the statistical parametric mapping (SPM) framework (see Materials and methods for further details). Models such as the two presented here could be generalized to all BCI applications and used to identify brain networks that encode specific behavioral and clinical target variables.

## Discussion

Bidirectional brain computer interfaces will revolutionize the treatment of previously intractable brain disorders with brain signal decoding based adaptive neuromodulation. DBS provides a unique platform to trailblaze neurophysiological approaches, disease-specific modulation and computational strategies for brain signal decoding for next-generation brain implants. Here, we investigated clinical and computational strategies for grip-force decoding as a representative and pathophysiologically relevant behavioral target variable. We used multimodal invasive neurophysiology time-series data in PD patients undergoing DBS electrode implantation. Our findings can be broken down into four advances to the field: (1) we developed a new decoding approach based on multispectral time-concatenated band-power measures, subjected to Bayesian optimized extreme gradient boosted

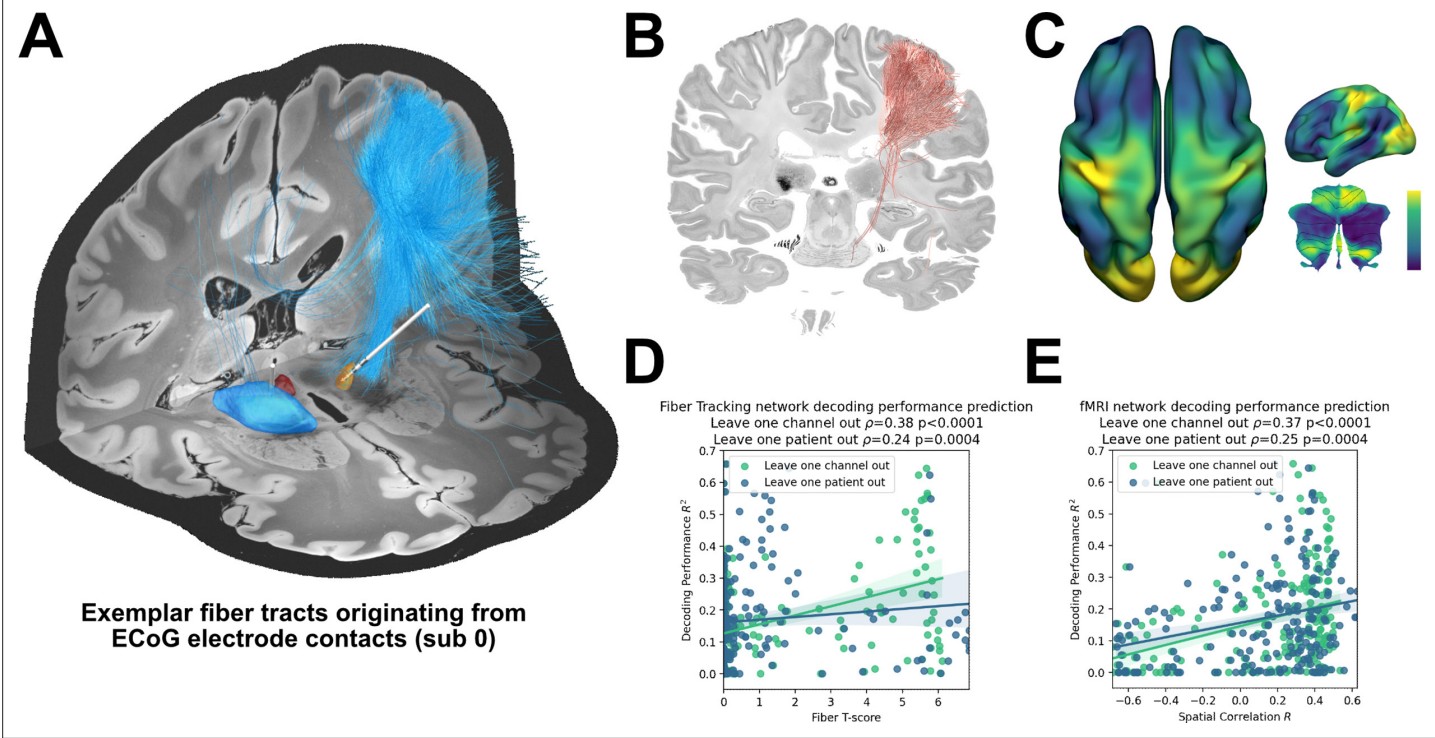

**Figure 7.** Structural and functional movement decoding network analysis reveals cerebellar as well as sensorimotor cortical decoding capacity. (**A**) Visualization of fibers originating from the ECoG recording locations of subject 1. (**B**) Decoding performance across all subjects and channels significant fiber tracts are displayed. All ECoG contacts were projected to the left hemisphere. For every fiber a t-test statistic between connected and unconnected brain regions was calculated. Only significant fibers, indicating structural connectivity to grip-force decoding performance, are shown. (**C**) The optimal R-Map is shown for the cortical surface as well as cerebellum for fMRI functional connectivity. Fingerprints were calculated between the functional connectivity of every electrode contact to all other voxels. The R-Map was then calculated as a correlation between individual contact fingerprints and the contact specific $R^2$ decoding performance. (**D**) Fiber tracking connectivity predicts grip-force decoding performance (leave one channel out cross validation $\rho$ =0.38, p<0.0001, leave one patient out cross validation $\rho$ =0.24, p=0.0004) (*Figure 7—source data 1*). Here, each individual point represents a statistic of connected and unconnected fibers of each contact or patient. The previously calculated fiber statistic within each cross-validation fold could thus predict the channel or patient specific performance. (**E**) Functional connectivity predicts decoding performance (leave one channel out cross validation $\rho$ =0.37, p<0.0001, leave one patient out cross validation $\rho$ =0.25, p=0.0004) (*Figure 7—source data 2*). The spatial correlation between individual fingerprints and the cross-validation specific R-Map, predicts left out decoding performances.

The online version of this article includes the following source data and figure supplement(s) for figure 7:

**Source data 1.** Fiber Tracking network decoding performance prediction.

**Source data 2.** fMRI network decoding performance prediction.

**Figure supplement 1.** 'Prediction Network Mapping' allows for prediction of machine learning decoding performances using functional and structural connectivity.

ensembles (XGBOOST): this outperformed traditional linear model-based methods and may be generalized to all brain signal-based regression problems. (2) Next, we demonstrate that electro-corticography signals outperform subthalamic LFP for grip-force decoding, supporting the utility of additional ECoG in adaptive DBS research for PD patients. (3) Our findings link PD motor impairment, PD pathophysiology with deterioration in decoding performance, highlighting a potential impairment in movement coding capacity through subthalamic low-beta bursts during motor preparation and execution periods. (4) Finally, we could significantly predict how well a specific recording site would perform to decode grip force based on brain connectivity. This novel framework (termed prediction network mapping) can be used in future implants to identify connectomic networks from which brain sensing can predict symptoms and behavior.

## Limitations

Our analysis is retrospective in nature and the data were obtained in context of a Go/No-Go task, which may have implications on the generalizability of the findings in the application during naturalistic

**Table 1.** Subject characteristics.

| N | Gender | UPDRS total | Hemisphere | Age | Movements | Disease duration [years] | ECoG Strip Contact Number Left | ECoG Strip Contact Number Right |
|---|--------|-------------|------------|-----|-----------|--------------------------|--------------------------------|---------------------------------|
| 0 | Male | 28 | R | 60.3 | 128 | 10.7 | 0 | 6 |
| 1 | Male | 27 | L+R | 51.2 | 464 | 14 | 28 | 28 |
| 2 | Male | 33 | L+R | 53.8 | 213 | 7.2 | 8 | 8 |
| 3 | Male | 31 | L+R | 44.2 | 285 | 10.1 | 8 | 8 |
| 4 | Male | 32 | 2L+2 R | 63.6 | 381 | 13.1 | 28+8 | 28+8 |
| 5 | Male | 52 | L | 59.6 | 84 | 5.9 | 6 | 0 |
| 6 | Male | 55 | L | 71.6 | 161 | 1.4 | 6 | 0 |
| 7 | Male | 50 | L | 52.5 | 131 | 8.7 | 6 | 0 |
| 8 | Male | 62 | L+R | 66.8 | 547 | 9.8 | 6 | 6 |
| 9 | Male | 48 | L | 67.9 | 86 | 17.1 | 6 | 0 |
| 10 | Female | 31 | R | 69 | 205 | 10.4 | 0 | 6 |

behavior. All model training and evaluations were conducted offline. Nevertheless, we took meticulous care to exclude any circularity in processing and machine learning applications. To this date, such circularities are overlooked in some movement decoding papers with filtering, normalization and time frequency transformation across entire sessions, thus reaching into the future from the point of the individually decoded sample. Ridding our analysis from data that would be unavailable in a real-time setting as reported in this study, leads to worse performances, but gives a more realistic estimate of model performance in the clinical use-case. While gripping is a relevant motor skill for human behavior, our findings are restricted to the decoding of grip-force and may have limited generalizability to other movements. The overall number of patients in this study is low. This may have limited a more detailed analysis of bias and other factors, beyond the described correlation of clinical symptom severity, subthalamic beta burst dynamics, electrode location and connectomics. Most importantly, the signal to noise ratio may further impact decoding accuracies differently for ECoG and LFP signals. This could in part explain why decoding from ECoG signals may benefit more from complex and non-linear model architectures. The comparability of ECoG and LFP recordings was further affected by the higher number of available ECoG channels, when compared to only three bipolar LFP channels. However, the large effect size of superior decoding performances with ECoG may indicate that this bias does not relevantly impact the interpretation of our findings. An additional limitation was the relatively small amount of available data per patient, which was constrained by the intraoperative setting (see *Table 1*). For deep learning approaches, we expect better performances with increased dataset sizes, which may become available, either through externalized extraoperative recordings (*He et al., 2021*) or sensing enabled implantable devices (*Opri et al., 2020*; *Gilron et al., 2021*). Importantly, our finding that decoding performances from single contacts outperform multi-electrode models may be a consequence of a combination of short recording durations in this study, suboptimal computational model selection and the fact that sensorimotor cortex and STN are part of the same circuit that is synchronized in oscillations. While we have made an effort to accommodate models that are optimized for spatio-spectral feature learning, and we are confident that these cannot outperform single channel approaches in this dataset, future studies should cautiously reinterrogate this issue in larger datasets, for example by implementing neural networks optimized for this purpose (*Peterson et al., 2021*). Finally, we should acknowledge that the exploration of the neural feature space in this study was non-exhaustive, and further raw data features, such as the local motor potential (*Mehring et al., 2004*), waveform shape features (*Cole and Voytek, 2017*), and aperiodic signal components (*Wilson et al., 2022*) could further improve decoding performances in future movement decoding studies.

## Decoding grip force based on invasive electrophysiology

Our study defines a novel computational strategy to decode grip-force based on ECoG and LFP in patients undergoing DBS for PD. It explores defined oscillatory feature sets and compares machine-learning models with varying complexity, from linear models to artificial neural networks and regression trees. ECoG-based movement decoding of varying movement types has been previously investigated in epilepsy patients that underwent electrophysiological monitoring (*Leuthardt et al., 2004*) through which local motor potentials and gamma band activity were highlighted as informative features (*Gunduz et al., 2016*). First analyses based on STN-LFPs in PD patients have shown that Wiener Filter architectures can be successfully used for grip-force decoding (*Tan et al., 2016*; *Shah et al., 2018*). The present study extends these previous reports to a continuous non-trial-based decoding approach. Furthermore, a direct comparison of ECoG and LFP performance with relation to systematic machine learning methods was lacking. Our findings indicate that sensorimotor ECoG recordings are more informative than LFP recordings from the STN for grip-force decoding. While this finding is robust, we should acknowledge that the size and shape of electrodes (see *Supplementary file 1a*) and the spatial orientation and size of the neural architectures that are sampled are not directly comparable across these methods. Thus, it is difficult to derive the relative importance of the different brain regions for grip-force and vigor processing in motor control from this comparison. Instead, we interpret our result as a practical demonstration of the greater utility of ECoG signals for movement decoding. The results in this study are based on extracted band-power features and show superior performances with XGBOOST, when compared to other model architectures and algorithms. More specifically, best performances were obtained for Bayesian optimized XGBOOST models trained on data from single ECoG channels without additional benefit from channel combinations or combined ECoG and STN channel sets. In the future, this machine learning approach can be adopted to extend the clinical utility of invasive brain stimulation approaches for other brain disorders, e.g. through decoding of tics or obsessive compulsive behavior in neuropsychiatric DBS indications.

## Towards machine-learning-based adaptive stimulation in Parkinson's disease

Adaptive DBS (aDBS) has the potential for significant innovation in movement disorders (*Starr, 2018*). For Parkinson's disease, different control policies of subthalamic beta band activity are now tested in clinical trials to improve the treatment for patients with akinetic rigid dominant PD (https://clinical-trials.gov/ Identifier: NCT04681534, NCT04547712) (*Little et al., 2013*; *Arlotti et al., 2018*; *Velisar et al., 2019*). Beyond subthalamic beta power, ECoG recordings were previously used to successfully decode the presence of dyskinesia through elevated levels of gamma band synchronization. This could be used to reduce stimulation intensity to alleviate medication and stimulation induced dyskinesia (*Swann et al., 2018*). Such single biomarker approaches have the advantage that pathophysiological mechanisms may be the direct target of intervention, while machine learning based decoding methods derive correlates of symptoms and behavior indirectly through learning potentially noisy correlations (*Neumann et al., 2019*). Therefore, single biomarker based aDBS presents an optimal starting point for investigating the clinical utility of aDBS in controlled study designs. However, single biomarkers alone cannot account for the diverse and complex set of clinical signs of PD and behavior, for example during gait (*Molina et al., 2021*; *Thenaisie et al., 2022*), speech, and tremor (*Hirschmann et al., 2013*; *Hirschmann et al., 2017*). Here, a versatile decoding based control algorithm may further improve clinical outcome for these patients in the future (*Neumann et al., 2019*; *Merk et al., 2022a*). Indeed, machine learning-based decoding has been successfully described in first translational breakthrough studies (*Opri et al., 2020*; *Gilron et al., 2021*; *He et al., 2021*). In a complementary approach, we focused on direct grip-force decoding, motivated by the hypothesis that future aDBS studies increasing DBS amplitude during periods of higher movement vigor may advance the successful treatment of bradykinesia in PD. While our previous findings indicate that relative amounts of beta can still signal bradykinesia during movement, (*Lofredi et al., 2019*; *Feldmann et al., 2021*) further positive control parameters could keep stimulation proportional to intended movement vigor. Moreover, recent reports that beta power correlates negatively with phasic dopamine release may further substantiate the idea of movement/kinematics based STN stimulation to support intrinsic movement related dopamine signals (*Schwerdt et al., 2020*). We may speculate that DBS constitutes a network modulation that is similar to dopamine transients by suppressing local

firing of the subthalamic nucleus (*Milosevic et al., 2018*) and shifting the balance of basal ganglia from indirect to direct pathway activity. As highlighted above it was recently shown in non-human primates that phasic decreases in beta in the basal ganglia are correlated to phasic dopamine signals during movement (*Schwerdt et al., 2020*). Thus, in order to support the intrinsic dopaminergic capacity of PD patients, future machine learning based aDBS approaches could be complemented by algorithms that inform the stimulation on behavioral and motor adjustments to mimic intrinsic phasic dopamine signals. Previous studies have successfully decoded the presence of movement using cortical beta activity (*Opri et al., 2020*) which could also become a viable treatment option in PD. However, getting an estimate of movement vigor that is through the prediction of grip-force may complement advanced aDBS control policies, as multivariate models emerge for the next-generation of neurotherapeutics.

Notably, the proposed adaptive stimulation would require a fast algorithmic adaptation of stimulation to ongoing behavior. This could be combined with additional slower adaptations in response to medication or sleep cycles. Specifically for PD, beta-activity-based adaptive stimulation can be well suited to track the patient's overall symptom state (*Tinkhauser and Moraud, 2021*) while more rapid stimulation adaptations based on vigor can follow fast kinematic changes. The utility of vigor-based stimulation and the combination of this approach with additional slower adaptation algorithms, require further proof-of-concept studies before the clinical utility can be foreseen. In our study, we demonstrate that motor symptom severity itself can have direct and negative effects on decoding performance, which we should keep in mind during clinical decision making. Previous studies have shown that the presence of beta bursts correlated with motor performance in cortex (*Little, 2019*) and STN (*Torrecillos et al., 2018*), which could degrade decoding performance (*Khawaldeh et al., 2020*). Our study replicates and extends these findings, as we show a direct correlation between movement related beta burst dynamics and PD motor sign severity. More importantly, our results show that the amount of time the STN is bursting in the low-beta band, during motor preparation and movement execution is inversely correlated with ECoG based grip-force decoding performance. An obvious interpretation of this finding is that excessive synchronization in the STN may impair flexible motor control by decreasing information coding capacity and neural entropy as previously suggested in animal studies (*Mallet et al., 2008*; *Cruz et al., 2009*) and recently suggested for subthalamic beta bursts (*Velasco et al., 2022*). Again based on the inverse relationship of beta activity and dopamine (*Schwerdt et al., 2020*), we may speculate that beta bursts may relate to transient dips in dopamine signaling. Dopamine was shown to precede and invigorate future movement (*da Silva et al., 2018*). If subthalamic beta bursts indicate phasic decreases in dopaminergic innervation, we could expect a loss of invigoration and reinforcement of ongoing neural population activity in the cortex – basal ganglia – thalamic loop, which offers an elegant explanation for the lower decoding performance from ECoG signals in the absence of obvious cortical activity patterns.

Beyond beta bursts our findings indicate general impact of motor symptoms in the hypodopaminergic state on machine learning based kinematic decoding capacity. This highlights the conceptual relevance of disease specific interactions with computational models. Interestingly, in the hypodopaminergic state, the model output underestimated the grip force extent produced by the patients. This could reflect a loss of neural vigor representations related to insufficient dopaminergic modulation (*Turner and Desmurget, 2010*). In the future, we will have to account for the individual impact of disease specific changes in brain signals that affect decoding performance. Further, our results corroborate the notion that dopamine plays a key role in coding and modulating neural representations of movement kinematics in the human brain.

## Connectomics can aid the discovery of brain networks underlying encoding of clinical and behavioral target variables

Decoding performance for clinical BCI may be drastically improved when adjusting brain signal recording sites to the underlying interconnected network that is relevant for encoding of the specific target behavior. For instance, when decoding language or speech, one could envision that recordings at either Broca's or Wernicke's region could be helpful, but a combination of both could be optimal. The two regions form a network with direct connections via the Arcuate Fascicle. In the present study, we have leveraged multisite recordings from various electrode locations across patients to identify the network that would be most informative for grip force decoding. For this endeavor, we adapted two existing methods that are able to isolate (i) connected voxels and (ii) connected fiber tracts (*Horn*

*et al., 2017b*; *Li et al., 2020*) associated with a specific target metric (such as grip-force decoding performance in the present case). While Euclidean distance to motor target, i.e. hand knob area for ECoG and therapeutic target for STN, was significantly correlated with decoding performance within-subject, this simplistic notion could not predict decoding performance across channels or patients. Thus, proximity to landmarks alone does not reliably help the identification of optimal recording sites. Given the complexity and vast distribution of movement related brain areas, from cerebellum to frontal cortex to parietal cortex, it may not be surprising that whole-brain connectomics outperform single region of interest based distance metrics for predicting informative recording locations. The development of a connectomic identification of optimal decoding locations has important implications in clinical adoptions of BCI technology. Preoperative identification of brain networks would allow the design of optimal electrode architectures and targeted implantation to cover strategic nodes of distributed networks for decoding of clinical variables and behavior. Moreover, connectomic approaches can inform the optimal spatial feature selection of pretrained machine learning models to facilitate brain signal decoding without the requirement for individual (re-)training. Importantly, the connectomic models that we used can be trained based on multiple dimensions of input-output relationships, for example for decoding of behavior like grip-force, but also for decoding clinical signs, such as tremor or mood disturbances. Thus, when implanting a high-density ECoG grid, connectomic analyses can generate target specific contact combinations, for example focusing on primary cortex for tremor and supplementary motor area for motor intention and bradykinesia. Our results highlight the utility of whole-brain connectomics to predict machine learning-based brain signal decoding performance that can be generalized to any bidirectional clinical brain-computer interface use-case. In the future, neurosurgeons may not target individual sensing locations in isolation, but instead determine optimal implant trajectories in accordance with whole-brain connectomic fingerprints for optimal BCI performance.

## Conclusion

Our analysis from PD patients undergoing DBS implantation showed that ECoG recordings outperform STN-LFP recordings for grip-force decoding throughout different machine learning methods, with XGBOOST showing the highest performance. Parkinsonian motor sign severity and subthalamic low-beta bursts were associated with loss of decoding performance, indicating a specific link between PD pathophysiology, kinematic coding capacity and motor impairment. To investigate the spatial relationship of ECoG decoding performances in the brain, we have formalized a connectomic framework that could cross-predict decoding performances across recording sites and patients, based on underlying whole brain MRI connectivity patterns. Our findings highlight the utility of ECoG for intelligent adaptive stimulation in PD, corroborate the role of PD symptom severity in kinematic coding and pave the way for connectomic neurosurgery for machine learning-based brain signal decoding. We hypothesize that future neurotechnological treatments may have the potential to outperform traditional drug regimes, due to a key advantage in the temporal and spatial precision of therapeutic delivery towards a precision medicine approach for intelligent adaptive DBS (*Neumann et al., 2019*; *Neumann and Rodriguez-Oroz, 2021*; *Merk et al., 2022a*).

## Materials and methods
### Participants

The current study is based on previously published data (*Alhourani et al., 2020*). In brief, subthalamic LFP and subdural ECoG recordings were simultaneously acquired from 11 PD patients. The patients were subjected to bilateral STN-DBS lead implantation, as proposed by standard clinical indications criteria. In accordance with protocol #PRO13110420, approved by the Institutional Review Board of the University of Pittsburgh, informed consent for all patients was obtained prior to any surgical procedure. The subject characteristics are detailed in *Table 1*. UPDRS Part III scores for the off-medication conditions were collected in a time period of 1–3 months prior to surgery by movement disorder neurologists. Dopaminergic medications were withheld for at least 12 hr before intraoperative testing.

## Behavioral paradigm

The behavioral task performed for this study was previously described (*Kondylis et al., 2016*; *Alhourani et al., 2020*; *Fischer et al., 2020*) and it is schematically shown in *Figure 1A*. The task included Go/No-Go cues with randomized inter-trial interval durations. Feedback durations were adjusted based on grip force reaction times. In the present analyses, time-series were virtually streamed as continuous data to simulate real-time grip-force decoding, irrespective of task trials. Subjects were fully awake, and no anesthetic agents were administered for at least 1 hr before the task procedure. No medication was given during the task. The task paradigm was implemented using the Psychophysics Toolbox (*Brainard, 1997*) on a portable computer. The trials consisted of a simultaneous presentation of a yellow traffic light in the center of a screen, and a cue on one side indicating which hand the subject should use for the subsequent response of squeezing the handgrip. The cue remained on screen for 1000–2000ms, followed by the traffic light changing either green or red, signaling a 'go cue' and 'no-go cue', respectively. Subjects performed the task for a cumulative total time of 10–25 min. As the present study focuses on grip-force decoding performance based on the electrophysiological signals, all sessions containing valid movements were merged per subject for further analysis. To validate that the used grip-force label in our data varies not only between two movement states, but constitutes a relevant regression problem with varying force amplitude and velocity, all movement maximum amplitudes and velocity traces are visualized in the *Figure 1—figure supplement 1*.

## Electrophysiological recordings

Subdural electrode strips were implanted temporarily through standard frontal burr holes located near the coronal suture and aimed posteriorly to the hand knob motor cortex region. Strip targeting has been previously described and was based on markings of stereotactically defined overlying scalp locations (*Kondylis et al., 2016*). STN-DBS electrodes were implanted bilaterally, targeting the dorsolateral motor area of the STN. ECoG data were recorded intra-operatively using six-contact (left n=5 patients, right n=3), eight-contact (left n=3, right n=3) and twenty-eight-contact (left n=2, right n=2) strip electrodes. The electrode details are shown in *Supplementary file 1a* and all ECoG and STN electrodes are plotted in *Figure 1B* (mean number of electrode contacts were 10.18±11.29 for left and 8.9±12 for right hemispheres). A referential montage was used in which the reference electrode was placed in the scalp and a ground electrode was placed in the skin overlying the acromion process. ECoG and STN signals were filtered (0.3–7.5 kHz), amplified, and digitized at 30 kHz using a Grapevine neural interface processor (Ripple Inc). Force signals were digitally recorded simultaneously with the ECoG and STN-LFP signals. LFPs from the STN were recorded using the clinical DBS lead (model 3389, Medtronic) from all four contacts and referenced offline in a bipolar montage. All signals were resampled to 1 kHz for offline analysis. To investigate the variability of grip-force as a potential bias for decoding performance, we calculated the variance of peak force across movement repetitions.

## Electrode localization

Subdural electrode reconstructions were obtained by aligning pre-operative MRI, intra-operative fluoroscopy, and postoperative CT. Representative images of this technique were previously shown in detail (*Randazzo et al., 2016*). In short, the CT and MRI were co-registered using mutual information using the SPM software library and rendered onto 3D skull and brain surfaces using Osirix (v7.5) (*Rosset et al., 2004*) and Freesurfer (v5.3) software packages (*Dale et al., 1999*), respectively. These surfaces and the fluoroscopy images were then aligned according to common points: stereotactic frame pins, implanted depth electrodes, and skull outline positions (*Randazzo et al., 2016*). The parallax effect of the fluoroscopic images was accounted for using the obtained distance from the radiation source to the subject's skull. Succeeding the surface-to-fluoroscopic image orientation alignment, a 3D location for each electrode contact was projected from the fluoroscopic image to the cortical surface. Deep brain stimulation electrode locations were reconstructed using the advanced neuroimaging pipeline defined by Lead-DBS using default settings (*Horn et al., 2019*). In brief, preoperative MRI and postoperative CT scans were co-registered and normalized to MNI 2009b NLIN ASYM space. Electrode artefacts were visually identified and marked to obtain MNI coordinates of DBS electrode contacts. All electrode localizations are visualized in *Figure 1B*.

## ECoG and LFP preprocessing and feature extraction

The entire preprocessing pipeline used in the present study was optimized for real-time performance and inspired by the Berlin Brain Computer Interface (*Blankertz, 2006*). Processing was performed in Python using custom code based on MNE-python (*Gramfort et al., 2013*), mne_bids (*Appelhoff et al., 2019*) and pybv (https://pybv.readthedocs.io/en/stable/). All raw data files were saved in the iEEG-BIDS structure (*Holdgraf et al., 2019*). To account for baseline drifts, the force traces were cleaned using a normalization approach presented for previous ECoG finger trajectory decoding (*Xie et al., 2018*). A real-time data stream of untouched electrophysiological raw data was emulated to ensure that all processing that can impact decoding is performed in a real-time compatible manner. Referencing was performed online (i.e. after streaming virtual data packets). All LFP recordings were referenced bipolarly, against the adjacent contacts (0–1, 1–2, 2–3 with contact 0 being the lowest by convention of the manufacturer). Throughout the manuscript, we adopt the clinical usage of electrodes (also named 'leads') and contacts from the DBS realm. During preprocessing (in pseudo real time), we derive three bipolar STN-LFP channels from four adjacent contacts in one DBS electrode (also called 'lead'). We also follow this nomenclature for ECoG, where we call the entire strip an 'electrode'. ECoG electrodes in our dataset can have varying number of contacts (see *Supplementary file 1a*). ECoG recordings were referenced by subtracting the common average of all ECoG electrodes, therefore the number of channels per ECoG electrode is equal to the number of contacts per strip. To facilitate computationally efficient real-time enabled algorithms, time frequency decomposition for the machine learning analysis was conducted by bandpass filtering in the $\theta$(4–8 Hz), $\alpha$(8–12 Hz), $\beta$(13–35 Hz), low $\beta$(13–20 Hz), high $\beta$(20–35 Hz), all $\gamma$(60–200 Hz), low $\gamma$(60–80 Hz) and high-frequency activity, (90–200 Hz) frequency bands. Overlapping broad $\beta$ and $\gamma$ bands were added in addition to subbands to enable the investigation of distinct interactions within these frequency bands (*Figure 1C*). To estimate band specific activity, different time durations were used for band-pass filtering with longer time segments for lower frequencies, and shorter time segments for higher frequencies ($\theta$=1000ms, $\alpha$ and $\beta$ bands = 500ms, $\gamma$ = 100ms). To get an estimate of amplitude of the activity in the filtered signals, variance was extracted in intervals of 1 s in a sliding window of 100ms resulting in a time resolution of 10 Hz. All variance estimates were normalized by subtracting and dividing by the median in a sliding window of 10 s to account for differences in impedance and proximity to the source before subjecting the data to the machine learning analysis. All features were clipped as an artifact rejection mechanism when they exceeded a normalized value of [–2 2]. The used normalization is fully compatible with a real-time prediction approach, as data acquired in the future do not influence the present predictions. See *Figure 1E* for an outline of the methods pipeline. For the purpose of visualization, Morlet wavelets (7 cycles) were used to demonstrate the entire time-frequency decomposition (*Figure 1C*).

## Machine learning training and evaluation

A rigorous nested cross-validation approach was implemented. An outer threefold cross validation split the data into folds of two third training and one third test set. For each individual channel, a Bayesian Optimization hyperparameter search (*Frazier, 2018*) was then conducted for 10 rounds using the training set only. For each round the training data was trained and tested in an inner threefold cross-validation with 80% training size. Post-hoc assessment confirmed convergence in performance after a maximum of 5 rounds in all recordings. The mean $R^2$ coefficient of determination of every test set estimate of the outer cross-validation was used as the performance measure as defined below:

$$R^2(y, \hat{y}) = 1 - \frac{\sum\limits_{i=1}^{n}(y_i - \hat{y}_i)^2}{\sum\limits_{i=1}^{n}(y_i - \bar{y})^2}$$

Since the $R^2$ metric can be lower than zero for predictions that are worse than constant predictions, we used a lower threshold at zero to make performances comparable for the purpose of visualization. The input features for every model were all eight previously described frequency bands. In order to test the contribution of time points preceding the decoded target sample, frequency band features of different time points were concatenated and compared with respect to their decoding performance. The present study investigated commonly used and promising linear and non-linear machine learning

algorithms, specifically elastic net regularized linear models, linear Wiener filters, neural networks, gradient boosted decision trees (XGBOOST) and source power comodulation.

## Linear models

Linear models can capture underlying feature dependencies and reveal those as correlations in each weight parameter. Input features are multiplied by a weight coefficient. The dot product of the weight vector $w$ and feature vector $x$ is then shifted by the bias $b$. The feature vector in this analysis is the vector of all frequency bands for a single time point. The prediction label $y$ is the baseline corrected gripping force. For a linear regression the activation function is linear, is defined as follows:

$$y = wx + b$$

To prevent overfitting, regularization in the form of $l_1$ and $l_2$ norm is commonly used. Here we tested different parameters of the elastic-net (enet) regularization (**Zou and Hastie, 2005**), which is a combination of the $l_1$ and $l_2$ norm specified by the regularization hyperparameters $\alpha$ and $\rho$, respectively. The objective function of the enet model follows:

$$\min_{w} \frac{1}{2n_{samples}} \|Xw - y\|_2^2 + \alpha\rho \|w\|_1 + \frac{\alpha(1-p)}{2} \|w\|_2^2$$

where $X$ is a matrix of dimension $n$ x $m$ whom i$^{th}$ row is the feature vector $x$ of size $m$ and $w$ is the solution vector, which, due to the $l_1$ sparse regularization term, most of the coefficient will be expected to be zero. For hyperparameter-search, $\alpha$ and $\rho$ were both sampled from a uniform distribution ranging from zero to one. Since elastic nets are solved using gradient descent, the maximum training iteration also needs to be specified. Here an iteration number of 1000 has been used. The implementation was done using the scikit learn Python package (**Pedregosa, 2011**).

## Wiener filters

Tan et al. described the use Wiener filters in the application of force estimation from STN-LFP signals (**Shah et al., 2018**). Here, the output $y$ is a weighted sum of features in the time and frequency domain in the weight matrix $W$. $I$ frequency band features are used together with $J$ lags. For the regression analysis the activation function is kept linear, as follows:

$$y(n) = \sum_{j=0}^{J} \sum_{i=0}^{I} w_{ij} x_i (n - j)$$

This equation has a closed form solution, known as the normal equation (**Proakis and Monolakis, 1996**). Wiener filters essentially implement a multivariable linear model with multiple time-steps. Using Wiener filters, we tested the contribution of different concatenated time-steps of brain signals preceding the decoded target sample. This provides insight about the optimal feature length in the time domain.

## Neural networks

We have further investigated the utility of artificial neural networks. While linear models and Wiener filters may underfit the data, neural networks can be very complex and have a higher risk to overfit with increasing complexity. The ideal model architecture finds a balance between under and overfitting to the training dataset. In this context not only single weight correlations of band features could contribute to force decoding performances, but a richer representation of feature invariances in combinations of different frequency bands may be learned by additional layers and units of the model. The architecture of neural networks is derived from linear models with non-linear activation functions, which are referred to in this context as units. Multiple units are combined in different layers with different activation functions.

Explicitly, the output $y$ of the i$^{th}$ unit in layer $l$ is the weighted sum of activations of the previous layer units $y_k^{l-1}$ with weights $w_{ik}^l$,

$$y_i^l = f^l \left( \sum_k w_{ik}^l y_k^{l-1} + b_i^l \right)$$

Neural networks are trained through a cost function using a gradient descent algorithm. Hyper-parameters were adjusted in order to prevent over -and underfitting (*Geman et al., 1992*). Here, neural networks were tested with at least one hidden layer. The input nodes of this layer were in the hyperparameter search uniformly sampled in a range of 1–10. The number of hidden dense layers were sampled from a range of 1–3 layers. The hidden dense layer neurons were uniformly sampled in a range of 1–10. Sigmoidal and hyperbolic tangent activation functions were tested in the hidden layers. After each hidden layer a batch normalization layer and a dropout layer with a factor of 0.2 was added. The output activation function was set linear. The used training algorithm was the Adam optimizer (the learning rate was sampled from a log uniform distribution from 0.0001 to 0.01, $\beta_1$ was set to 0.9, $\beta_2$ to 0.999 and $\varepsilon$ to 0.999). The Adam optimizer improves backpropagation such that each weight parameter is adapted according to its first and second momentum (*Kingma and Ba, 2015*). Each neural network was trained using 1,000 epochs with a batch size of 100. The loss function was set to the mean squared error. To prevent overfitting, the training set was further split into train and validation set with 80% train. The validation dataset was then used for early stopping with a patience parameter of 10 epochs. The model with lowest validation error is then used for test set prediction. Due to poor performances, the inner cross-validation was left out for the neural network training sequence. Neural Networks were implemented using the TensorFlow framework (*Abadi, 2016*).

## Gradient boosted trees using the XGBOOST framework

A common problem with neural networks is the high dependency on the provided set of features and potential to learn spurious input-output associations. In this analysis, a feature vector of all 8 frequency bands concatenated for 5 time points requires a Wiener Filter with 40 weights. In an architecture-like neural networks all these features are contributing to the overall force prediction, nevertheless not all weight parameters are promising. Decision Tree algorithms overcome this problem naturally by implementing optimization of input feature use in their architecture. Thus, decision trees and random forests, first described by *Breiman, 2001*, were proven to be a robust, accurate and successful tool for solving machine learning tasks, including classification, regression, density estimation and manifold learning or semi-supervised learning (*Gall and Lempitsky, 2013*). Random forests are an ensemble method consisting of many decision trees. A decision tree is a statistical optimal data segregation method, that is only controlled by conditional sequences. Different implementations were proposed on top of Decision Trees. AdaBoost (*Schapire, 2009*) is an adaptive learning algorithm that builds up successive decision trees iteratively. By that an ensemble of multiple weighted weak learners are combined to yield a strong estimator. Gradient Boosting is built using the same concept. According to Empirical Risk Minimization it fits each decision tree based on the residuals of a defined objective function. This objective function is typically based on an error loss and a regularization term. The model is initialized using a constant value. In an iterative process, the new trees are added to the model up till the maximum defined estimators are reached. Here, the scalable tree boosting framework XGBOOST (*Chen and Guestrin, 2016*) was used. In this analysis the number of boosting rounds is set to 10. The depth of each tree is sampled uniformly in a range from 1 to 100. When adding new trees to the model the parameter learning rate $\eta$ is scaling the contribution of each tree prediction and is sampled here log uniformly from of the range [ $10^{-5}$ , 1]. Regularization in Gradient Boosted Trees is controlled by different factors. One of the factors is the minimum splitting loss $\gamma$. For every decision tree, new nodes were added only if the *gain* metric was above $\gamma$. It is here sampled from a uniform distribution between 1 and 10. Hyperparameters for all used machine learning methods are listed in detail in *Supplementary file 1b*.

## Source power comodulation

A state-of-the-art movement prediction approach is the source separating framework called Source Power Comodulation (SPoC) (*Dähne et al., 2014*). Oscillatory sources are here extracted based on their power comodulation with the force gripping target. SPoC was implemented using the MNE framework (*Gramfort et al., 2013*). Thus, discriminant neural sources are made visible. In this context, the band-power at each frequency band of interest was calculated by taking the logarithm of the variance of the projected signal in the source space. For sake of comparison, only one spatial filter was used for feature computation at each frequency band. In the same manner as before, a Wiener filter was then applied in order to resample time lags up to 500ms. Here again, the band power

features are then used as input features. A Bayesian Optimization hyperparameter search was also here implemented for both the enet model as well as the XGBOOST framework with the aforementioned parameters.

## Hyperparameter search: Bayesian optimization

All models underwent an extensive hyperparameter search using Bayesian optimization. A common problem using machine-learning algorithms is finding the optimal hyperparameter settings given a certain architecture. *Grid search* exhaustively tries out all provided hyperparameters while *Random search* only draws random parameters from the given hyperparameter distributions. Sampling the error loss function can be computationally expensive. Bayesian Optimization formulates this problem into an optimization problem. Here, a cost function is minimized given a set of hyperparameters. Instead of sampling from the objective cost function, a probabilistic model is defined. The hyperparameters minimizing the negative expected improvement *are selected* given a multinomial Gaussian process *using a Matern kernel*. Those parameters are then used to sample from the respective regressor in the given dataset. The resulting error is used to update the gaussian process distribution and given the maximum expected improvement, the next best hyperparameter set is drawn. This process is repeated for the elastic net, *neural networks and* XGBOOST architecture for 10 iterations. For every round, a threefold cross validation is used in order to prevent overfitting. Given log-uniform distributions a wide range of hyperparameters can thus be sampled in a computationally efficient manner. The implementation was done using the scikit-optimize framework (https://scikit-optimize.github.io/stable/). *Supplementary file 1b* lists the hyperparameters subjected to Bayesian optimization. The chosen methodology is non-exhaustive and primarily serves the comparison of variance in decoding explained by the recording location of the signal (ECoG vs. STN), motor symptom severity (UPDRS-III), beta bursts and brain networks. It further gives an intuition about the potential of more complex and elaborate machine learning methods for brain computer interfaces.

## Definition of best model and best channels

Previous studies have repeatedly demonstrated that using a single optimal channel in the STN is advantageous over using all available channels (*Shah et al., 2018*). Most importantly, addition of more channels leads to decreased generalization and higher risk of overfitting with little performance benefit. Based on these results and to account for varying numbers of available electrode contacts, one channel with optimal decoding performance on the cross-validation test set was chosen per patient to quantify and compare decoding performance for the ECoG and STN analysis across patients. Since hyperparameter optimization is implemented only within each inner cross validation fold, any circularity and data leakage is circumvented. A robust decoding performance estimate is thus obtained through left out testing data only.

## Analysis of beta bursts during motor preparation and movement execution periods

To investigate a potential relationship between grip-force decoding performance and beta burst activity, we have adopted a previously validated approach to movement related burst analyses (*Torrecillos et al., 2018*; *Lofredi et al., 2019*). Therefore, the beta feature time-series were used and a threshold constituting the 75th percentile of the rest periods were calculated. Next, threshold crossings of at least 100ms lengths in the motor preparation (−1–0 s with respect to movement) and movement execution (0–1 s with respect to movement execution) were marked as bursts. In previous reports, the most informative metric was the 'time spent in burst' which is calculated as the sum of burst durations in the time period of interest. This metric is directly proportional to the burst probability at a given time-point. All burst analyses were repeated for the low-beta and high-beta bands in ECoG and STN-LFP. The times spent in bursts were correlated with UPDRS-III and ECoG based decoding performances.

## Prediction Network Mapping with whole-brain connectomics

To investigate whether decoding performance from different recording locations can cross-predict decoding performances across patients, we developed a whole-brain connectomics based approach. Therefore, ECoG electrode recording locations were projected to normative structural and functional

MRI data (Parkinson's Progression Markers Initiative [PPMI]; https://www.ppmi-info.org/) using Lead-DBS software in Matlab (https://www.lead-dbs.org/). (*Horn et al., 2019*) The PPMI connectomes of patients with PD (n = 74) was priorly computed (*Ewert et al., 2018*) and has been used in context of DBS multiple times (*Horn et al., 2017c*; *Neumann et al., 2018*; *de Almeida Marcelino et al., 2019*; *Lofredi et al., 2021*). No patient-specific diffusion or functional MRI was required for this analysis. Seeding from each recording site resulted in connectivity profiles (fingerprints) that were expressed as voxel-wise whole-brain volumes for functional and structural connectivity and a set of streamline connections for structural connectivity. We have adapted three previously published methods leveraging normative connectomes as predictive models.

First, fiber streamlines representative of structural connectivity between ECoG channels and all other brain areas were isolated and assigned with a 'Fiber T-score', associating XGBOOST decoding performance with the fiber tracts connectivity from respective ECoG recording locations across patients using mass-univariate two-sample t-tests between $R^2$ scores in connected vs. unconnected recording locations. Only fibers with significant t-scores surviving FDR correction at an alpha level 0.05 were considered further. Next, T-values were used as weights in an aggregated fiber score to predict out of training sample channel and patients' performances. Next, functional connectivity maps were used to generate an 'R-Map', a connectivity model which is associated with optimal decoding performance, by performing voxel-wise correlations of connectivity and decoding performance from recording locations. The connectomic fingerprint from each recording location can then be assigned a spatial correlation coefficient that may have predictive value for the underlying decoding performance. The predictive value of these two methods were confirmed using 'leave-one-channel-out' and 'leave-one-subject-out' cross-validation. Finally, statistical parametric mapping was used to confirm the described correlations of structural and functional connectivity using linear-mixed effects models. In a voxel-wise approach, structural connectivity between ECoG channels and all other brain areas was calculated using Lead Mapper (https://www.lead-dbs.org/). Statistical voxel-wise correlation between decoding performance and structural and functional connectivity, separate mixed effects models, with a subject based random effect, were corrected for multiple comparisons with random field theory as implemented in the Statistical parametric mapping (SPM12) toolbox (https://www.fil.ion.ucl.ac.uk/spm/). Functional connectivity strengths between recording sites and sensorimotor cortex (peak coordinate x = –38, y = –22, z=72), parietal lobe (x=6, y = –32, z=82), striatum (x = –34, y = –24, z=26). and cerebellum (x=18, y = –50, z = –50 and x = –22, y = –52, z = –54) accounted for decoding performance. Similarly, for structural connectivity, a significant cluster in the sensorimotor region (x = –44, y = –18, z=70) correlated with high decoding performance. All connectivity analyses were performed using ECoG recording locations with contralateral $R^2$ performances (*Figure 1E*). A schematic illustrating the different steps of functional and structural prediction network mapping can be found in *Figure 7—figure supplement 1*.

## Statistical analysis

Results are stated as mean ± standard deviation. All significance testing was performed using two-sided Monte-Carlo permutation tests and bootstrapping. p-Values were obtained by shuffling value positions and determining the resulting original rho value percentile in the distribution of surrogate combinations. Spearman's correlations were performed because of small sample size and varying distributions. Clinical correlations were performed using preoperative UPDRS-III total scores. To test for the temporal specificity of the clinical correlation with decoding performance, we performed sample-wise correlations of decoding output with UPDRS-III total scores across subjects. Multiple comparisons were corrected by adjusting the significance threshold α to the false discovery rate (*Benjamini and Hochberg, 2016*).

## Data availability

All raw data in BIDS for iEEG format are openly available through the Harvard Dataverse (https://doi.org/10.7910/DVN/IO2FLM). Code and derived source data for the reproduction of all figures and the machine learning and statistical analysis are provided through GitHub (https://github.com/neuromodulation/ECoG_vs_STN; *Merk, 2022b*; copy archived at swh:1:rev:09d3ea5b846681d28edb26943b4315ae5f5a37dd).

## Acknowledgements

The present manuscript was supported through a US-German Collaborative Research in Computational Neuroscience (CRCNS) grant to RST, RMR and WJN with funding from the German Federal Ministry for Research and Education and NIH (R01NS110424). Further funding was provided through Deutsche Forschungsgemeinschaft (DFG, German Research Foundation) – Project-ID 424778371 – TRR 295 to AH and WJN AH was supported by the German Research Foundation (Deutsche Forschungsgemeinschaft, Emmy Noether Stipend 410169619 and 424778371 – TRR 295) as well as Deutsches Zentrum für Luft- und Raumfahrt (DynaSti grant within the EU Joint Programme Neurodegenerative Disease Research, JPND). Some data used in the preparation of this article were obtained from the PPMI database (https://www.ppmi-info.org/). For up-to-date information on the study, visit https://www.ppmi-info.org/. PPMI, a public–private partnership, is funded by the Michael J Fox Foundation for Parkinson's Research. For funding partners, see https://www.ppmi-info.org/fundingpartners.

## Additional information

### Funding

| Funder | Grant reference number | Author |
| --- | --- | --- |
| National Institutes of Health | R01NS110424 | Robert S Turner<br>Robert Mark Richardson<br>Wolf-Julian Neumann |
| Bundesministerium für Bildung und Forschung | FKZ01GQ1802 | Wolf-Julian Neumann |
| Deutsche Forschungsgemeinschaft | 410169619 | Andreas Horn |
| Deutsche Forschungsgemeinschaft | Project-ID 424778381 – TRR 295 | Andreas Horn<br>Wolf-Julian Neumann |

The funders had no role in study design, data collection and interpretation, or the decision to submit the work for publication.

### Author contributions

Timon Merk, Conceptualization, Formal analysis, Funding acquisition, Methodology, Software, Visualization, Writing – original draft, Writing – review and editing; Victoria Peterson, Formal analysis, Methodology, Software, Writing – review and editing; Witold J Lipski, Investigation, Methodology, Writing – review and editing; Benjamin Blankertz, Funding acquisition, Writing – review and editing; Robert S Turner, Funding acquisition, Supervision, Writing – review and editing; Ningfei Li, Software, Writing – review and editing; Andreas Horn, Methodology, Resources, Software, Writing – review and editing; Robert Mark Richardson, Conceptualization, Data curation, Funding acquisition, Investigation, Methodology, Project administration, Resources, Supervision, Writing – review and editing; Wolf-Julian Neumann, Conceptualization, Funding acquisition, Investigation, Methodology, Project administration, Resources, Software, Supervision, Validation, Visualization, Writing – review and editing

### Author ORCIDs

Timon Merk http://orcid.org/0000-0003-3011-2612
Witold J Lipski http://orcid.org/0000-0003-1499-6569
Benjamin Blankertz http://orcid.org/0000-0002-2437-4846
Robert S Turner http://orcid.org/0000-0002-6074-4365
Andreas Horn http://orcid.org/0000-0002-0695-6025
Robert Mark Richardson http://orcid.org/0000-0003-2620-7387
Wolf-Julian Neumann http://orcid.org/0000-0002-6758-9708

### Ethics

Human subjects: The patients were subjected to bilateral STN DBS lead implantation, as proposed by standard clinical indications criteria. In accordance with protocol #PRO13110420, approved by

the Institutional Review Board of the University of Pittsburgh, informed consent for all patients was obtained prior to any surgical procedure.

## Decision letter and Author response
Decision letter https://doi.org/10.7554/eLife.75126.sa1
Author response https://doi.org/10.7554/eLife.75126.sa2

---

## Additional files

### Supplementary files
• Supplementary file 1. Electrode details, Bayesian optmization hyperparemeters and best subject performances. (a) Electrode Details (b) Bayesian Optimization Hyperparameters (c) Best channel $R^2$ performances.

• Transparent reporting form

### Data availability
All raw data in BIDS for iEEG format are openly available through the Harvard Dataverse (https://doi.org/10.7910/DVN/IO2FLM). Code and derived source data for the reproduction of all figures and the machine learning and statistical analysis are provided through GitHub (https://github.com/neuromodulation/ECoG_vs_STN, copy archived at swh:1:rev:09d3ea5b846681d28edb26943b4315ae5f5a37dd).

The following dataset was generated:

| Author(s) | Year | Dataset title | Dataset URL | Database and Identifier |
|---|---|---|---|---|
| Merk T, Peterson V, Lipski W, Blankertz B, Turner RS, Li N, Horn A, Richardson RM, Neumann WJ | 2022 | Replication Data for: Electrocorticography is superior to subthalamic local field potentials for movement decoding in Parkinson's disease | https://doi.org/10.7910/DVN/IO2FLM | Harvard Dataverse, 10.7910/DVN/IO2FLM |

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
