## [Editor Report]

This paper evaluates movement decoding from intracranial brain recordings in patients with Parkinson's disease. Interestingly, the authors demonstrate that cortical recordings (electrocorticography) outperform subthalamic nucleus in decoding grip force. This work will be of interest to brain computer interface, movement disorder, motor control, and general neurophysiology communities.

---

## [Decision Letter]

**Decision letter after peer review:**

Thank you for submitting your article "Electrocorticography is superior to subthalamic local field potentials for movement decoding in Parkinson's disease" for consideration by *eLife*. Your article has been reviewed by 3 peer reviewers, and the evaluation has been overseen by a Reviewing Editor and Michael Frank as the Senior Editor. The following individual involved in review of your submission has agreed to reveal their identity: Kevin Wilkins (Reviewer #1).

Essential revisions:

1) The paper reports best performance with single channel ECoG inputs and no additional benefit from channel combinations or combined ECoG and STN channel sets (summarized from lines 322-324, and shown in Figure 4). This result requires further discussion – does it truly suggest that using a single channel is optimal (as seems to be suggested by the authors) or does it rather speak to a need for models which more appropriately combine information from separate sources? While these specific XGBOOST models may have failed to benefit from larger feature combinations, the usefulness and generalizability of that knowledge is limited without more exhaustive testing and discussion. Theoretically, any new features added to an existing model should be capable of benefiting the combined model, provided those features both contain some amount of information about the predicted variable and are being optimally integrated. This is related to the topics of ensemble learning, boosting, and regularization, which are a part of the current paper. We recommend the authors further develop this aspect.

2) A primary weakness of the paper is that some of the results and ideas are underdeveloped. While identifying a relationship between motor symptom severity and decoder performance is interesting on its own, there is no further investigation of this result for providing better intuition about why the relationship exists and what can or should be done about it. The authors discuss the potential impact β bursts might have on decoder performance but do not pursue any analysis of their own data to determine whether those proposed hypotheses are supported. We encourage the authors to explore this space a little more either with additional analyses or at least additional discussion. For instance, the authors report a relationship between UPDRS scores and decoding performance, which they suggest may be valuable to clinical decision making and may be explained by an association between β bursts and motor performance (summarized from lines 367-372). Elaborating on these points would be of great value to the manuscript. The authors could assess whether β bursts are necessary for their observed relationship between decoding performance and UPDRS scores by removing β features from their decoders and re-testing (however, this would only test for direct effects that β-band activity makes as a regressor). Additionally, if β bursts are necessary for producing this observation, are there any adjustments to the models that could accommodate the changes in how β-band activity is temporally distributed? Finally, a more thorough description of how this relationship would be valuable to clinical decision making would be quite nice. Whether in terms of computational modelling, dopamine replacement therapy, or other routes – it would benefit the reader to have some commentary on precisely how this might change the trajectory of a patient's treatment.

3) A wide-range of decoding performance is seen across participants with a group of good performers and a group of low-performers. The findings from the secondary analyses on impairment levels and electrode location/connectivity may explain some of these differences, but it is unclear to what extent or if other factors are at play. Further, the cohort is notably small, especially considering the heterogeneity of electrode location. Although this does not limit the major finding of the superiority of sensorimotor ECoG over STN signals, it does limit the ability to understand the observed individual differences in decoding performance. These limitations should be noted/discussed.

4) When comparing cortical and subthalamic electrodes the size and structure of the probe may be different. This means that instead of comparing apples to apples, it is more like comparing apples to oranges. This does not completely undermine the result because the difference in decoding between the areas, even given experimental differences, is likely to be of interest to clinical researchers studying DBS. If the surface area of the electrode is different between the two regions, then this could be a factor in decoding performance that does not have to do with brain region. Additionally, the electrodes in the subthalamus nucleus are circular, which are likely targeting very different neural populations across the probe within the small nucleus, which is different from the cortical electrodes which are on the surface targeting neural populations which are adjacent. Both of these factors (e.g. size and shape) could contribute to differences in decoding performance regardless of brain region. We did not see details of the electrodes in the method section, but this would be important to report as surface area is related to the number of neurons/dendrites summing to create the LFP, and this might lead to qualitatively different results for something like hand gripping irrespective of area. Similarly, with the shape of the electrode. These details will be an important addition to the paper and something that others can continue to investigate (e.g., researchers who have different size or shape of electrodes in the STN). We are sympathetic that this is not a variable that the researchers can change given the clinical nature of DBS, but the surface area of electrodes in each area should be mentioned in the method section, and if the surface area of the electrodes are different, then it should also be mentioned as a limitation in the Discussion section.

5) We wonder whether you have fully explored the neural feature space. One suggestion is that you also include the time domain data as a feature along with your frequency bands. Some papers have shown pretty good decoding with this feature – sometimes called the local motor potential. Here are some papers which discuss this feature in more detail. This could be an interesting addition especially if it performs well as it requires little preprocessing for studies doing online preprocessing and decoding.

Flint, R. D., Wang, P. T., Wright, Z. A., King, C. E., Krucoff, M. O., Schuele, S. U.,.… & Slutzky, M. W. (2014). Extracting kinetic information from human motor cortical signals. Neuroimage, 101, 695-703.

Mehring, C., Nawrot, M. P., de Oliveira, S. C., Vaadia, E., Schulze-Bonhage, A., Aertsen, A., & Ball, T. (2004). Comparing information about arm movement direction in single channels of local and epicortical field potentials from monkey and human motor cortex. Journal of Physiology-Paris, 98(4-6), 498-506.

Schalk, G., Kubanek, J., Miller, K. J., Anderson, N. R., Leuthardt, E. C., Ojemann, J. G.,.… & Wolpaw, J. R. (2007). Decoding two-dimensional movement trajectories using electrocorticographic signals in humans. Journal of neural engineering, 4(3), 264.

6) With respect to Figure 2, given how similar the descriptive power plots are, we were surprised that low γ has much larger weights compared to high γ or HFA. It looks like you aren't using regularization for your linear regression model. If your features (band pass filters) are highly correlated, the interpretation of the weights might not be meaningful. Have you thought about using ridge regression or lasso to deal with your seemingly highly correlated features? If not, then I don't believe it makes sense to try and interpret the weights. It looks like you do use regularized regression later but looking at the method section for your linear regression model there is no regularization term – so based on that it seems like for this first section it is just standard linear regression. We would suggest also using regularized regression for these analyses as interpreting the weights of linear regression with highly correlated features may be problematic.

7) The R2 plots all appear to have a lower limit of 0. This, combined with some of the descriptions in the Methods, suggest that these performance metrics are the squared correlation coefficient. The computational form of the coefficient of determination R2, which is 1 – sum[(y_true – y_pred)^2] / sum[(y_true – y_mean)^2], is a more appropriate metric to use for evaluating decoder performance in cross-validated settings. The squared correlation coefficient is not equivalent in this case, as illustrated by the fact that it cannot be negative. Using the computational form of R2 allows the reader to readily compare predictions against a naïve estimator that predicts the sample mean – if the decoder being evaluated has an R2 below zero, then it is worse than simply predicting the mean. The correlation coefficient is also invariant to scale and translation, which are important features of decoder performance that are captured by the computational form of R2. If in fact the computational form of R2 has been used, please explicitly state that it has been (by simply including the equation in methods), since the results presented in the paper do appear to show otherwise.

8) We had a difficult time understanding the logic and the methods of your last section which relates decoding performance with connectivity maps. For example, after reading the methods section, I was still unclear how you determined if a fiber was significant or not. I believe that this section needs more detail and clarity. For example, you have analyses for structural and functional connectivity, but for the functional connectivity we could not find anything in the method section about what the patient was doing when this was computed – were the patients at rest, were they doing the same gripping task? These details are important for understanding the analyses and interpretation.

9) Similarly, the prediction network mapping methods would be more clear if represented schematically and/or with accompanying equations. These methods were very difficult to understand given the current descriptions in the Results text and Methods.

10) The authors are unable to fully rule-out that the superiority of ECoG signals is not dependent on the task performed (i.e., grip force). The authors claim that the findings support the utility of additional ECoG in adaptive DBS research for PD patients. Although it is certainly expected that sensorimotor ECoG would provide a richer signal compared to STN LFPs due to both signal size, complexity, and relation to movement, the data in this paper is inherently limited to the grip task performed. This limitation should be noted/discussed.

11) The results of the paper are all centered on decoding grip-force. The authors suggest that this decoding could be used as a control signal for adaptive deep brain stimulation in patients with Parkinson's disease, increasing stimulation when higher movement "vigor" is predicted (summarized from lines 346-348). It seems that this would entail a fast algorithmic approach to adapting stimulation, making adjustments in response to frequently changing movement states. A comparison to different algorithmic approaches (for example, tracking slower fluctuations in medication state) would be helpful to have in the Discussion.

12) A Go/No-Go task is also used in the paper. This is potentially problematic, given that there may be modulation in the neural recordings associated with response inhibition. While the decoders are trained only to predict actual force production, it is possible that inhibitory responses could influence the decoder readout and impact the performance metrics on which every analysis of the paper hinges. It is also plausible that such inhibitory responses would asymmetrically affect the subthalamic decoders and the cortical decoders. Comparison of those two signal locations is the main topic of the paper, so this is a critical aspect to consider and discuss.

13) The results identify a stark difference in the way model "complexity" impacts decoding in the STN and cortex, yet this concept is not well-developed or discussed. What does this say about the nature of the information content in each area such that allowing for non-linear relationships benefits decoders based on one area, but harms decoders based on the other? Please discuss further.

14) The Bayesian Optimization hyperparameter search was run for a fixed number of iterations according to line 637. Were enough iterations used to reach convergence in performance? This would be a useful supplementary analysis.

15) In line 648, it is stated that a single best channel was chosen for each patient and brain area (STN, cortex) in part to "account for varying numbers of available electrode contacts". There seem to be far more ECoG channels than STN channels – does this introduce a bias in the results when taking the "best" performing channel from a larger candidate pool of ECoG channels as compared to the STN? The authors should elaborate on this and make any necessary corrections for bias.

16) The significantly connected fibers are shown in Figure 7B. However, due to the small nature of this sub-figure and only 1 slice shown, it is difficult to fully understand where these fibers tend to go (appears to be a combination of M1 and SMA/PM?). It may be helpful to describe in the text of the results where these fibers are going. For instance, Figure 7C gives a great look into the visualization of the topography of decoding performance. Although that type of figure may not be possible with the fiber tract analysis, at least a text description would be helpful for the reader and allow further interpretation of those results.

17) The supplementary figure provides transparency regarding the variability of observed grip performances. One remaining question was whether the observed variability in decoding performance across individuals and the observed correlation between decoding performance and impairment level may actually be related to the observed variability in grip force production. Do the more impaired individuals display greater variability across trials in grip force production compared to those with less impairment? And is this then creating a more difficult decoding problem for those individuals with more extreme variability compared to individuals who have consistent force production curves?

---

## [Author Response]

Essential revisions:1) The paper reports best performance with single channel ECoG inputs and no additional benefit from channel combinations or combined ECoG and STN channel sets (summarized from lines 322-324, and shown in Figure 4). This result requires further discussion – does it truly suggest that using a single channel is optimal (as seems to be suggested by the authors) or does it rather speak to a need for models which more appropriately combine information from separate sources? While these specific XGBOOST models may have failed to benefit from larger feature combinations, the usefulness and generalizability of that knowledge is limited without more exhaustive testing and discussion. Theoretically, any new features added to an existing model should be capable of benefiting the combined model, provided those features both contain some amount of information about the predicted variable and are being optimally integrated. This is related to the topics of ensemble learning, boosting, and regularization, which are a part of the current paper. We recommend the authors further develop this aspect.

We thank the reviewers for raising this important point. The fact that single channels outperform channel combinations has led to extensive discussion among the study team over the course of the analyses. We agree with the reviewers that the finding is counterintuitive from a theoretical perspective. Therefore, we have spent significant time and effort into understanding this specific issue and have trained models that make optimal use of both spatial and spectral information. Our results showed lower performances of combined sources compared to single contact models. To test for combined models of multiple contacts, we tested the Source Power Comodulation (SPoC) model by Dähne et al. 2014 https://doi.org/10.1016/j.neuroimage.2013.07.079. This method utilizes spatial information and has been widely used in EEG regression analysis. We trained linear models as well as XGBOOST decoders on top of SPoC extracted features to account for linear and non-linear interactions. But both models were outperformed by XGBOOST using a single best cross validated contact (see Figure 3 and Supplementary File 1c for decoding performance comparisons). We hypothesize that the small dataset size in the intraoperative recording setting is a limiting factor to generalize performances of multichannel models. We are confident that our findings are robust for the present dataset but believe that extending this section of the article with the limited data available is beyond the interest of the general readership of *eLife*. We conclude that there are two primary reasons for this finding, (a) the limited amount of data at hand, that is inherently restricted by the intraoperative recording setting, which is decisive for the risk of overfitting more complex models, and (b) the fact that the brain signals that are most informative for grip-force decoding all originate from the same cortex-basal ganglia-thalamic circuit. The positive side of this finding is that the channel count of emerging neurotechnology is often defined and limited by hardware manufacturers, and it appears that even devices with limited channel numbers may have a potential for good decoding performances. Nevertheless, we agree that our manuscript with the presented data cannot extrapolate towards all potential settings for invasive movement decoding and therefore discuss this topic more cautiously. Finally, we currently work on the implementation of convolutional and recurrent neural networks for movement decoding similar to Peterson et al., 2021 https://doi.org/10.1088/1741-2552/abda0b but our preliminary decoding performanes remain below the presented results in this paper and require further optimization and validation.

The limitations section of the discussion now reads:

“An additional limitation was the relatively small amount of available data per patient, which was constrained by the intraoperative setting (see Table 1). For deep learning approaches we expect better performances with increased dataset sizes, which may become available, either through externalized extra operative recordings (He et al., 2021) or sensing enabled implantable devices (Opri et al., 2020; Gilron et al., 2021). Importantly, our finding that decoding performances from single contacts outperform multi-electrode models may be a consequence of a combination of short recording durations in this study, suboptimal computational model selection and the fact that sensorimotor cortex and STN are part of the same circuit that is synchronized in oscillations. While we have made an effort to accommodate models that are optimized for spatio-spectral feature learning, and we are confident that these cannot outperform single channel approaches in this dataset, future studies should cautiously reinterrogate this issue in larger datasets, e.g. by implementing neural networks optimized for this purpose (Peterson et al., 2021).”

2) A primary weakness of the paper is that some of the results and ideas are underdeveloped. While identifying a relationship between motor symptom severity and decoder performance is interesting on its own, there is no further investigation of this result for providing better intuition about why the relationship exists and what can or should be done about it. The authors discuss the potential impact β bursts might have on decoder performance but do not pursue any analysis of their own data to determine whether those proposed hypotheses are supported. We encourage the authors to explore this space a little more either with additional analyses or at least additional discussion. For instance, the authors report a relationship between UPDRS scores and decoding performance, which they suggest may be valuable to clinical decision making and may be explained by an association between β bursts and motor performance (summarized from lines 367-372). Elaborating on these points would be of great value to the manuscript. The authors could assess whether β bursts are necessary for their observed relationship between decoding performance and UPDRS scores by removing β features from their decoders and re-testing (however, this would only test for direct effects that β-band activity makes as a regressor). Additionally, if β bursts are necessary for producing this observation, are there any adjustments to the models that could accommodate the changes in how β-band activity is temporally distributed? Finally, a more thorough description of how this relationship would be valuable to clinical decision making would be quite nice. Whether in terms of computational modelling, dopamine replacement therapy, or other routes – it would benefit the reader to have some commentary on precisely how this might change the trajectory of a patient's treatment.

This is a very relevant point, and we agree with the reviewer that the provided insight in the initial version of the paper was limited. Therefore, we have significantly extended the analyses on this topic for the revision. Before expanding on the new analyses, we would like to highlight some caveats when discussing the relationship of pathological β oscillations and movement decoding that β activity is one of the most potent features for prediction of movement. Also, when visualizing coefficients from linear models, we observe that strong β desynchronization is an important feature for movement decoding. Consequently, β power in the resting period is a requirement for the desynchronization, which means that removing these features from a model is expected to significantly deteriorate decoding performance. As by suggestion of the reviewer we have now analysed the performance changes by removing β activity from ECoG and STN decoding performances as shown in Author response image 1.

**Author response image 1. sa2fig1:** 

This should remind us of the fact that β activity per se is a physiological signal and it can be challenging to differentiate pathological and physiological aspects of it. Therefore, we have tried to conceptualize a new analysis that (a) abstains from manipulation of model features because of the direct and often hardly interpretable effects on decoding performance, (b) provides pathophysiological insight into PD and (c) is embedded into a physiologically plausible line of reasoning from previous literature. Upon reflection on these points, we realized that we could investigate the relationship of subthalamic β burst dynamics on ECoG based grip-force decoding performance. This is relevantly embedded in previous works, that have demonstrated that “time spent in β burst” in the subthalamic nucleus can be associated with worsening of movement performance (Torrecillos et al., 2018, J Neurosci, https://doi.org/10.1523/jneurosci.1314-18.2018). The “time spent in burst” metric was used in various previous publications on the relationship of β bursts and motor performance beyond the abovementioned one (Lofredi et al., 2019, Tinkhauser et al., 2020, Khawaldeh et al., 2020) and essentially reflects the likelihood of β bursts occurring in the STN during movement preparation and/or performance. It is practical for this purpose as it is easy to compute and sensitive even during periods of low average β activity even when the threshold is defined by the resting β measures. Note that all of the previous studies were restricted to subthalamic LFP recordings and none had the opportunity to compare burst dynamics with ECoG signals or decoding. Thus, we have adopted the methodology of these studies and analyzed the impact of β bursts on ECoG decoding performance, by identifying threshold crossings above the 75^th^ percentile of the resting low-β power feature vector of at least 100 ms length. We have focused on the low-β (13-20 Hz) band, because it was shown to be the primary relevant frequency band in our previous works (Lofredi et al., 2019) but also compared the correlation to the high-β frequency band (20-35 Hz) further in Author response image 2. We found a significant negative correlation of ECoG decoding performance (best performance with XGBOOST as highlighted in the manuscript before) and STN burst dynamics (time spent in burst as explained above and reported in previous studies). Specifically, we found robust significant correlations for STN time spent in β burst during motor preparation (Rho = -0.76, P = 0.001; -1 s to 0 s with relation to movement onset) and during movement execution (Rho = -0.71, P = 0.008; 0 s to 1 s after movement onset).

Importantly, the correlations were specific to the STN and did not yield significant results for bursts in cortex (all P>0.05). To uncover a potential relationship of PD symptom severity and burst measures we calculated correlations with UPDRS-III scores and found significant correlations for time spent in burst for STN during motor preparation (Rho = 0.63, P = 0.02) and movement (Rho = 0.56 P = 0.05) but not for time spent in burst from ECoG activity (P>0.05).Interestingly, the negative correlation of time spent in burst with motor preparation was specific for the low-β band and not significant for the high-β band (p>0.05), while the correlation during movement was significant but lower than for low β (Rho = -0.71 p = 0.007). Again, correlations of time spent in burst in the high-β frequency band in cortex with decoding performance were not significant (p>0.05) for neither motor preparation nor movement phase.

We believe that these findings add significant value to the manuscript, and we are truly thankful to the reviewers who have motivated us to dive deeper into this analysis. We have thus decided to merge previous figures 3 and 4 and include an additional figure (now figure 5) and added the required methods and results to the manuscript as will be highlighted below. As always with interesting scientific findings, they spark new questions, e.g. the question whether β bursts can impair corticosubthalamic communication, e.g. by suppressing neural phase locking of subthalamic neurons to cortical γ (see Fischer et al., 2020 *eLife* https://elifesciences.org/articles/51956) or modulating coherence and granger causality. Finally, the question arises whether dopaminergic medication suppresses β bursts and improves movement decoding. We will address these relevant questions in future studies for which we are now collecting data from externalized leads in Berlin, but unfortunately recruitment is going slow during the pandemic.

Finally, we want to further address the reviewers comments with respect to a discussion on potential practical implications. Foremost, we want to reinstate our original argument that our findings should spark a general awareness that the pathophysiology of brain disorders will have significant impact on machine learning models for clinical brain computer interfaces, beyond obvious / direct changes in the input features themselves. This general statement will be important beyond deep brain stimulation and could also hold value for first application of other brain computer interfaces, such as Neuralink or similar.

In the particular case of Parkinson’s disease and grip-force decoding, we demonstrate that subthalamic but not cortical β burst synchronization even before movement execution is associated with deterioration of decoding performance from cortex. An obvious interpretation of this finding is that excessive synchronization in the STN may impair flexible motor control by decreasing information coding capacity and neural entropy as previously suggested in animal studies (Mallet et al. 2008, https://doi.org/10.1523/JNEUROSCI.0123-08.2008; Cruz et al., 2009 https://doi.org/10.1152/jn.00344.2009) and recently suggested for subthalamic β bursts (Velasco et al. 2022, https://doi.org/10.1109/TBME.2022.3142716).

More generally, we speculate that β activity can be anticorrelated with phasic dopamine release (Schwerdt et al., 2020 https://doi.org/10.1126/sciadv.abb9226). If this holds true also for the STN, we may associate β bursts with phasic dips in dopamine signalling. Dopamine was shown to precede and invigorate future movement (da Silva et al., 2018; https://doi.org/10.1038/nature25457). If subthalamic β bursts indicate phasic decreases in dopaminergic innervation, we could expect a loss of invigoration and reinforcement of ongoing neural population activity in the cortex – basal ganglia – thalamic loop, which offers an elegant explanation for the lower decoding performance from ECoG signals in the absence of pathological cortical activity patterns. In the future, this could be further investigated through direct recordings of cortical neural population activity with Neuropixels that we and our collaborators have previously implanted in first human cases during epilepsy surgery (Paulk et al., 2022; https://doi.org/10.1038/s41593-021-00997-0) optimally coupled with electrochemical sensing of dopamine (Schwerdt et al., 2018; https://doi.org/10.1038/s42003-018-0147-y) when it becomes available for human application. On a more short-term perspective, we may investigate changes in patterning and amplitude of γ band activity with dopaminergic medication during externalized recordings. We are currently working on this and found preliminary unpublished evidence that cortical γ power is increased with dopaminergic medication when compared to OFF medication, which resembles previous work shown for the STN (Lofredi et al., 2018, https://doi.org/10.7554/*eLife*.31895).

While the practical clinical requirement for brain signal decoding beyond pathological signals remains distant, we finally want to state shortly, how we believe our results could affect clinical decision making: In general, our findings may suggest that patients with earlier disease stages or lower symptom severity may be better suited for clinical brain computer interfaces that rely on movement decoding. Another intuitive conclusion would be that decoding performance can be improved with dopaminergic medication. However, we should be cautious with this speculation, as the UPDRS-III in the OFF state reflects concurrent clinical state severity but is also correlated with chronic dopaminergic degeneration. Thus, neurodegeneration could lead to chronic loss of cortical information coding capacity that is not reversible with medication. We will address this important question in future studies from externalized recordings. To compensate, computational methods to improve decoding performance could account for presence of β bursts and increase model output in an adaptive way. More work is required to conceptualize this further, but we believe that such ideas could be interesting to follow up on. We hope that the editorial team recognizes our appreciation of the discussion. We have now substantially revised abstract, methods, results and discussion to accommodate the new analyses and have included a new Figure 5.

Abstract:

“ECoG based decoding performance negatively correlated with motor impairment, which could be attributed to subthalamic β bursts in the motor preparation and movement period. This highlights the impact of PD pathophysiology on the neural capacity to encode movement kinematics.”

Results:

“To better understand the relationship of PD pathophysiology and grip-force decoding performance we have further investigated associations between cortical and subthalamic β burst dynamics. We follow the methodology of previous reports that demonstrated that the time spent in β burst correlates with impairment of movement kinematics (Torrecillos et al., 2018). Β bursts were defined as threshold crossings of the β feature vector above the 75^th^ percentile of the baseline period. Following the previous finding that specifically the time-spent in low-β but not high-β bursts was correlated with PD motor impairment (Lofredi et al., 2019), we investigated these bands separately for the motor preparation period (-1 to 0 s with respect to movement onset) and movement execution period (0 to 1 s following movement onset). To uncover a potential relationship of the β-burst metric with PD pathophysiology, we performed correlations with UPDRS-III total scores. Significant correlations were found between UPDRS-III and low-β bursts in STN-LFP signals during motor preparation (ρ = 0.63, p = 0.02; Figure 5A) and movement execution (ρ = 0.56, p = 0.04; data not shown), but not for the high-β band (p>0.05). Conversely, for ECoG highbeta but not low-β burst dynamics during motor preparation but not movement periods were significantly correlated with UPDRS-III total scores (ρ = 0.55, p = 0.04). In summary, we provide evidence that both subthalamic and cortical β burst dynamics relate to PD motor sign severity with subthalamic low-β bursts showing the most robust correlations, both during motor preparation and movement periods.

To relate these findings to movement decoding performance from cortex, we correlated the grand average XGBOOST grip-force decoding performances from ECoG channels (as above for UPDRS-III) with high- and low-β burst dynamics in both ECoG and STN-LFP signals. ECoG based grip-force decoding performance was significantly correlated with subthalamic low-β burst dynamics during motor preparation (ρ = -0.76, p = 0.004) and movement execution (ρ = -0.71, p = 0.005; Figure 5B). Subthalamic burst dynamics in the high-β band also correlated with ECoG decoding performances during movement (ρ = 0.71, p = 0.007) but not motor preparation. Cortical burst dynamics from ECoG signals did not reveal significant correlations with ECoG based grip-force decoding performances. Relevant correlations alongside exemplar burst visualizations and corresponding grip-force decoding traces are shown in Figure 5.”

Discussion:

Our study replicates and extends these findings, as we show a direct correlation between movement related β burst dynamics and PD motor sign severity. More importantly, our results show that the amount of time the STN is bursting in the low-β band, during motor preparation and movement execution is inversely correlated with ECoG based grip-force decoding performance. An obvious interpretation of this finding is that excessive synchronization in the STN may impair flexible motor control by decreasing information coding capacity and neural entropy as previously suggested in animal studies (Mallet et al., 2008; Cruz et al., 2009) and recently suggested for subthalamic β bursts (Velasco et al., 2022). Again, based on the inverse relationship of β activity and dopamine (Schwerdt et al., 2020), we may speculate that β bursts may relate to transient dips in dopamine signaling. Dopamine was shown to precede and invigorate future movement (da Silva et al., 2018). If subthalamic β bursts indicate phasic decreases in dopaminergic innervation, we could expect a loss of invigoration and reinforcement of ongoing neural population activity in the cortex – basal ganglia – thalamic loop, which offers an elegant explanation for the lower decoding performance from ECoG signals in the absence of obvious cortical activity patterns.

Methods:

Analysis of β bursts during motor preparation and movement execution periods.

To investigate a potential relationship between grip-force decoding performance and β burst activity, we have adopted a previously validated approach to movement related burst analyses (Torrecillos et al., 2018; Lofredi et al., 2019). Therefore, the β feature time-series were used and a threshold constituting the 75th percentile of the rest periods were calculated. Next, threshold crossings of at least 100 ms lengths in the motor preparation (-1 to 0 s with respect to movement) and movement execution (0 to 1 s with respect to movement execution) were marked as bursts. In previous reports, the most informative metric was the “time spent in burst” which is calculated as the sum of burst durations in the time period of interest. This metric is directly proportional to the burst probability at a given time-point. All burst analyses were repeated for the low-β and high-β bands in ECoG and STN-LFP. The times spent in bursts were correlated with UPDRS-III and ECoG based decoding performances.

3) A wide-range of decoding performance is seen across participants with a group of good performers and a group of low-performers. The findings from the secondary analyses on impairment levels and electrode location/connectivity may explain some of these differences, but it is unclear to what extent or if other factors are at play. Further, the cohort is notably small, especially considering the heterogeneity of electrode location. Although this does not limit the major finding of the superiority of sensorimotor ECoG over STN signals, it does limit the ability to understand the observed individual differences in decoding performance. These limitations should be noted/discussed.

We fully agree with the reviewers. Although we believe we have further addressed this issue we are constantly dissatisfied with one source of additional bias, which is signal to noise ratio. Signal to noise ratio in (intracranial) EEG can be measured by assessing the presence of periodic activity over the 1/f aperiodic component or over the amplifier noise floor. The issue we have with this approach is that again, the presence of these activity patterns are direct features to our models and their relevance can be indirectly assessed through visualization of the correlation coefficients shown in Figure 2 of the original submission. On the other hand, presence of β activity can be affected by Parkinson’s disease and again it can be hard to disentangle the relationship of these activity patterns with physiology and pathophysiology of movement in an isolated way. Therefore, we have made an alternative effort in addition to the reflection on disease severity above and functional and structural connectivity profiles (Figure 7), we investigate an alternative measure of physiological signal to noise ratio. In brief, we hypothesize that higher noise contaminated signals contain higher amounts of uncorrelated components. In turn, less noisy signals contain most of the explained variance in fewer uncorrelated components. We used Principal Component Analysis (PCA) to project rest data to n uncorrelated components, where n is the number of features. We demonstrate that an increased amount of information content in higher order components is associated with lower decoding performance.

**Author response image 3. sa2fig3:** 

This may indirectly indicate that detrimental effect of signal noise. Upon reflection and in conjunction with our response to comment 2 however, it still remains unclear whether this increased “noise” reflects pathophysiology or differences in noise contamination from external signals. Therefore, we refrain from adding this to the manuscript. Instead, we highlight the limited amount of data and the potential impact of signal to noise ratio on decoding performance in the limitations section of the discussion, which now reads:“The overall number of patients in this study is low. This may have limited a more detailed analysis of bias and other factors, beyond the described correlation of clinical symptom severity, subthalamic β burst dynamics, electrode location and connectomics. Most importantly, the signal to noise ratio may further impact decoding accuracies differently for ECoG and LFP signals.”

4) When comparing cortical and subthalamic electrodes the size and structure of the probe may be different. This means that instead of comparing apples to apples, it is more like comparing apples to oranges. This does not completely undermine the result because the difference in decoding between the areas, even given experimental differences, is likely to be of interest to clinical researchers studying DBS. If the surface area of the electrode is different between the two regions, then this could be a factor in decoding performance that does not have to do with brain region. Additionally, the electrodes in the subthalamus nucleus are circular, which are likely targeting very different neural populations across the probe within the small nucleus, which is different from the cortical electrodes which are on the surface targeting neural populations which are adjacent. Both of these factors (e.g. size and shape) could contribute to differences in decoding performance regardless of brain region. We did not see details of the electrodes in the method section, but this would be important to report as surface area is related to the number of neurons/dendrites summing to create the LFP, and this might lead to qualitatively different results for something like hand gripping irrespective of area. Similarly, with the shape of the electrode. These details will be an important addition to the paper and something that others can continue to investigate (e.g., researchers who have different size or shape of electrodes in the STN). We are sympathetic that this is not a variable that the researchers can change given the clinical nature of DBS, but the surface area of electrodes in each area should be mentioned in the method section, and if the surface area of the electrodes are different, then it should also be mentioned as a limitation in the Discussion section.

We follow the line of argumentation regarding the lack of comparability of ECoG and LFP signals. In fact, we do not necessarily believe and did not state that the subthalamic nucleus does not encode grip-force or vigor signals or does this less than cortex. We have previously demonstrated that movement velocity scales with subthalamic γ band activity (Lofredi et al., 2018, https://doi.org/10.7554/*eLife*.31895). Thus, our assumptions in fact are quite the opposite and follows the vigorous tutor paradigm (Turner & Desmurget 2010; https://doi.org/10.1016/j.conb.2010.08.022). Thus, the comparison of ECoG and LFP signals was more practically motivated, to highlight the utility of ECoG in brain computer interfaces research in movement disorders and justify its use for further research. While widely adopted in the US, our own group is the first to implant ECoG electrodes in PD patients in Europe. There is ethical concern regarding safety (which was recently addressed by us in a retrospective analysis on 367 patients (Sisterson et al., 2021; https://doi.org/10.1093/neuros/nyaa592)) and practical concern regarding its utility which we clearly demonstrate in this paper. We now acknowledge the different nature of ECoG and LFP signals with respect to hardware in the discussion.

It now reads:

“Our findings indicate that sensorimotor ECoG recordings are more informative than LFP recordings from the STN for grip-force decoding. While this finding is robust, we should acknowledge that the size and shape of electrodes (see Supplementary File 1a) and the spatial orientation and size of the neural architectures that are sampled are not directly comparable across these methods. Thus, it is difficult to derive the relative importance of the different brain regions for grip-force and vigor processing in motor control from this comparison. Instead, we interpret our result as a practical demonstration of the greater utility of ECoG signals for movement decoding.”

Furthermore, beyond the differences in shape of ECoG vs. DBS electrodes, we realized that we further did not sufficiently highlight differences in shape of ECoG strip designs across patients. Size, shape and contact area of cortical strip and DBS electrodes are now listed in the Supplementary File 1a where we have added a table with these details:

5) We wonder whether you have fully explored the neural feature space. One suggestion is that you also include the time domain data as a feature along with your frequency bands. Some papers have shown pretty good decoding with this feature – sometimes called the local motor potential. Here are some papers which discuss this feature in more detail. This could be an interesting addition especially if it performs well as it requires little preprocessing for studies doing online preprocessing and decoding.Flint, R. D., Wang, P. T., Wright, Z. A., King, C. E., Krucoff, M. O., Schuele, S. U.,.… & Slutzky, M. W. (2014). Extracting kinetic information from human motor cortical signals. Neuroimage, 101, 695-703.Mehring, C., Nawrot, M. P., de Oliveira, S. C., Vaadia, E., Schulze-Bonhage, A., Aertsen, A., & Ball, T. (2004). Comparing information about arm movement direction in single channels of local and epicortical field potentials from monkey and human motor cortex. Journal of Physiology-Paris, 98(4-6), 498-506.Schalk, G., Kubanek, J., Miller, K. J., Anderson, N. R., Leuthardt, E. C., Ojemann, J. G.,.… & Wolpaw, J. R. (2007). Decoding two-dimensional movement trajectories using electrocorticographic signals in humans. Journal of neural engineering, 4(3), 264.

The reason we excluded raw signals was our main background in oscillations research where we typically disregarded any activity below 3 Hz as potentially contaminated by movement artifacts in these movement disorders patients. For the sake of expectation management we would like to state first, that we have decided not to include the local motor potential in the main manuscript, because changing the preprocessing and raw data feature set would affect every single result of the paper, which was beyond of what we could accomplish in a revision. Nevertheless, we realized that excluding that feature was indeed a loss, as we have run some tests with XGBOOST (for pragmatic reasons without Bayesian optimization leading to slightly lower performances overall) and found a relevant increase in decoding performance from inclusion of the local motor potential.

**Author response image 4. sa2fig4:** 

While this difference in our exemplary analysis was not significant, we will now add the local motor potential in all future movement decoding applications and we are thankful for the suggestion to asses it. We further highlight its potential in the limitations section of the discussion and use this to further mention other features that we are exploring in the development of our decoding toolbox (https://github.com/neuromodulation/py_neuromodulation):“Finally, we should acknowledge that the exploration of the neural feature space in this study was non-exhaustive, and further raw data features, such as the local motor potential (Mehring et al., 2004), waveform shape features (Cole and Voytek, 2017) and aperiodic signal components (Wilson, Castanheira and Baillet, 2022) could further improve decoding performances in future movement decoding studies.”

6) With respect to Figure 2, given how similar the descriptive power plots are, we were surprised that low γ has much larger weights compared to high γ or HFA. It looks like you aren't using regularization for your linear regression model. If your features (band pass filters) are highly correlated, the interpretation of the weights might not be meaningful. Have you thought about using ridge regression or lasso to deal with your seemingly highly correlated features? If not, then I don't believe it makes sense to try and interpret the weights. It looks like you do use regularized regression later but looking at the method section for your linear regression model there is no regularization term – so based on that it seems like for this first section it is just standard linear regression. We would suggest also using regularized regression for these analyses as interpreting the weights of linear regression with highly correlated features may be problematic.

We thank the reviewers for this important point. We realize that our figure was not explained clearly enough in the figure legend. Therefore, we would like to start with a more thorough explanation of the figure itself before replying specifically to the point with respect to regularization. First, we would like to reconcile the frequency bands: Low γ (60-80 Hz), HFA (90-200 Hz) and all γ (60-200 Hz). Figure 2A shows the model coefficients of the multivariable ordinary least squares linear regression model without regularization. Figure 2B in contrast shows coefficients from multiple univariable models that were trained separately with just one frequency band and time-point with respect to current decoded sample (not movement onset). We have done that to allow a direct comparison of individual features without interactions. We interpret that instantaneous low γ activity has the strongest individual positive association and instantaneous high-β has the strongest negative association with grip-force.

To address the comment about regularization we have further tested the following steps to investigate weight changes:

We retrained linear models similar to Figure 1A but added LASSO regularization and optimized the α regularization parameter, such that slight performance deterioration occurs. With this method only relevant features would remain in the regularized model, and others get penalized towards zero.

The top row of Author response image 5 demonstrates that regularization did not qualitatively affect coefficients. The lower left panel shows that the performance with regularization deteriorated. The lower right panel shows the individual data points and box plots of univariate models from single frequency band features similar to the x-axis time-point 0 in the original Figure 2B. These additional analyses corroborate our initial finding that indeed low γ (60-80 Hz) activity can have the strongest positive association with grip-force.We would also like to note that this figure was solely serving the purpose of feature visualization. The comparison of machine learning model performances in Figure 3 an Elastic Net Regression Model was chosen, which contains LASSO and RIDGE regression parameters that were obtained using a Bayesian Optimization hyperparameter search.

**Author response image 5. sa2fig5:** 

We now further clarify this issue in the figure legend of Figure 2, which now reads:

“Figure 2: Linear Models and Wiener Filters reveal temporally and spectrally specific coefficient distributions with grip-force decoding performance gain by including signals preceding the target sample by up to 500 ms. Multivariable linear model coefficients trained only from the instantaneous sample (0 time lag with respect to decoded target sample) including all frequency bands from best channels per patient resemble movement induced spectral changes with β desynchronization and γ synchronization (A). ECoG derived coefficients yield higher absolute values than STN-LFP derived coefficients. (B) Univariate frequency and time lag specific Linear Models were trained and visualized to improve interpretability of average coefficients in the absence of interactions. Low γ (60 – 80 Hz), HFA (90 – 200 Hz) and all γ (60 – 200 Hz) bands show stronger positive associations for contralateral over ipsilateral movements. Moreover, stronger associations are visible for ECoG over STN-LFP signals for β, HFA and γ bands. (C) Wiener Filters can integrate multiple time-steps in Linear Models leading to an incremental performance gain when signals are included preceding the current target sample by up to 500 ms**.**”

7) The R2 plots all appear to have a lower limit of 0. This, combined with some of the descriptions in the Methods, suggest that these performance metrics are the squared correlation coefficient. The computational form of the coefficient of determination R2, which is 1 – sum[(y_true – y_pred)^2] / sum[(y_true – y_mean)^2], is a more appropriate metric to use for evaluating decoder performance in cross-validated settings. The squared correlation coefficient is not equivalent in this case, as illustrated by the fact that it cannot be negative. Using the computational form of R2 allows the reader to readily compare predictions against a naïve estimator that predicts the sample mean – if the decoder being evaluated has an R2 below zero, then it is worse than simply predicting the mean. The correlation coefficient is also invariant to scale and translation, which are important features of decoder performance that are captured by the computational form of R2. If in fact the computational form of R2 has been used, please explicitly state that it has been (by simply including the equation in methods), since the results presented in the paper do appear to show otherwise.

We thank the reviewers for this important note. The computational form of the coefficient of determination was calculated, but clipped at 0 for the purpose of axis standardization. We further clarify this in the methods section, which now reads:

“The mean R^2^ coefficient of determination of every test set estimate of the outer cross validation was used as the performance measure as defined below:

R2(y,y^)=1−∑i=1n(yi−y⏞i)2∑i=1n(yi−y¯)2Since the R² metric can be lower than zero for prediction that are worse than constant predictions, we used a lower threshold at zero to make performances comparable for visualization purpose of visualizations.”

8) We had a difficult time understanding the logic and the methods of your last section which relates decoding performance with connectivity maps. For example, after reading the methods section, I was still unclear how you determined if a fiber was significant or not. I believe that this section needs more detail and clarity. For example, you have analyses for structural and functional connectivity, but for the functional connectivity we could not find anything in the method section about what the patient was doing when this was computed – were the patients at rest, were they doing the same gripping task? These details are important for understanding the analyses and interpretation.

Before diving into the details of this analysis, we would like to state that the used methods all rely on normative connectomes, in this case from a cohort of patients with Parkinson’s disease outside of the study sample. Thus, no patient individual diffusion or functional MRI was required. The methods are relatively new and have been developed for the use of clinical outcome prediction for deep brain stimulation network targets and were derived from Mike Fox’s approach to lesion network mapping (Fox MD, 2018; https://doi.org/10.1056/nejmra1706158). The idea is to use a normative connectome as a wiring diagram to reveal the shared network of distributed locations in the brain. This has led to numerous high-impact publications on the relevance of brain networks for neural disorders and neuromodulation targets, e.g. for a recent update in the field of deep brain stimulation see Hollunder et al., 2022 https://doi.org/10.1016/j.pneurobio.2021.102211. The present manuscript represents the first use of this method in the context of brain signal decoding. In this context we named it “Prediction Network Mapping” to indicate that we are using the network to estimate the predictive performance of a recording location. Thus, generally speaking, it allows the identification of a shared network connecting recording locations associated with good decoding performance. Specifically, in the present use case, it reveals the whole-brain network that connects ECoG recording locations that had good decoding performances. Even though in the case of movement decoding the result may look trivial, the method itself has impactful implications, as we demonstrate that it can be used to perform out-of-sample predictions on the expected decoding performance of a certain brain location. We are currently validating this method for across-patient brain signal decoding, where the right contact can be selected just based on the underlying network architecture of its location in the brain. We believe that this method will soon be adopted for other decoding purposes and believe it is a relevant novelty factor in this paper. The specific fiber tract method mentioned in the reviewers comment is named fiber filtering or discriminative tractography is a relatively new addition to the connectomics toolkit reported in Li et al., 2021 https://doi.org/10.1038/s41467-020-16734-3 to determine connectivity associated with response to DBS for OCD. Results are obtained by calculating the tract models from all ECoG recording locations and generate coefficients termed fiber T scores, that indicate the associated the association of the target variable for the recording location (in this case gripforce decoding performance) with the tract across all recording locations that share the tract. The relevant fiber bundle across all training subjects is determined by mass univariate Ttests FDR corrected for multiple comparisons. If this bundle is determined in a training group, its predictive performance can be validated on a test case through leave-one-subject or leave-one-electrode out cross-validation to predict decoding performance in an out-of-sample fashion, solely based on recording locations.

After rereading the paragraph we felt that it will be hard to explain the method more thoroughly without significantly extending the methods section. Given that the described methods are popular among DBS neuroimaging enthusiasts and are now frequently used by many groups and because they are not central to the present manuscript, we have decided to keep the methods section concise but add an explanatory schematic in response to comment 9 and a more clear statement regarding the absence of patient individual diffusion and functional MRI:

“To investigate whether decoding performance from different recording locations can cross predict across patients, we developed a whole-brain connectomics based approach. Therefore, ECoG electrode recording locations were projected to normative structural and functional MRI data (Parkinson's Progression Markers Initiative [PPMI]; www.ppmi-info.org) using Lead-DBS software in Matlab (www.lead-dbs.org).(Horn et al., 2019) The PPMI connectomes of patients with PD (n = 74) was priorly computed (Ewert et al., 2018) and has been used in context of DBS multiple times (de Almeida Marcelino et al., 2019; Horn et al., 2017b; Lofredi et al., 2021; Neumann et al., 2018). No patient specific diffusion or functional MRI was required for this analysis.”

9) Similarly, the prediction network mapping methods would be more clear if represented schematically and/or with accompanying equations. These methods were very difficult to understand given the current descriptions in the Results text and Methods.

We agree with the reviewers that the methods used for the connectomics analysis are quite dense and benefit from a schematic, which we have now added as an additional supplementary figure to the manuscript:

In the methods section we reference to the Figure:

“A schematic illustrating the different steps of functional and structural prediction network mapping can be found in Figure 7—figure supplement 1.”

10) The authors are unable to fully rule-out that the superiority of ECoG signals is not dependent on the task performed (i.e., grip force). The authors claim that the findings support the utility of additional ECoG in adaptive DBS research for PD patients. Although it is certainly expected that sensorimotor ECoG would provide a richer signal compared to STN LFPs due to both signal size, complexity, and relation to movement, the data in this paper is inherently limited to the grip task performed. This limitation should be noted/discussed.

We agree that the data presented is only limited to the single movement type of gripforce. And we expect the reviewers point would hold true for any kind of task, as the human ability to move is virtually infinite and can never be fully captured by repetition of certain aspects of movement. We now acknowledge the fact that our findings are not generalizable to movement per se, but are limited to grip-force in the limitations section of the discussion, which now reads:

“While gripping is a relevant motor skill for human behavior, our findings are restricted to the decoding of grip-force and may have limited generalizability to other movements.”

Nevertheless, to convince the reviewers that the findings may have some general worth in the context of movement we have performed further analyses on a dataset from an additional PD patient recruited at the Charité in Berlin, Germany. This subject was recorded with a cortical strip electrode (N=6 contacts) covering the sensorimotor cortex targeted as in the present cohort from Pittsburgh, as well as segmented DBS electrodes (N=8 contacts) in the STN. Here we recorded movements with a rotational handle, previously described by Lofredi et al. 2019, Neurobiology of Disease https://doi.org/10.1016/j.nbd.2019.03.013. The task instruction was to perform a voluntary self-paced rotational hand movement at an interval of approximately 10 s. The XGBOOST validation pipeline was the same as in the present manuscript. The target variable in this case was not grip force, but rotation amplitude.

**Author response image 6. sa2fig6:** 

We observe excellent decoding performances that are higher than on average than in the present manuscript. But more importantly, ECoG contacts are still outperforming recordings from the STN. In fact, the worst ECoG contact showed higher performances than the best STN contact, suggesting that the obtained grip force decoding results may hold at least in one other movement modality. Finally, we observe very similar results for a button press task in a different cohort from collaborators in Beijing. From our perspective, the question of specificity may be more relevant than the generalizability. Here, we believe that the primary source of specificity can be obtained through a denser coverage of somatotopy with higher density strips, e.g. as used in Chrabaszcz et al., 2019 https://doi.org/10.1523/jneurosci.284218.2019 where we showed lip vs. tongue specific activity during speech production. With low density strips with large surface areas and low impedances and with common DBS electrodes, we believe that the main signal that we can decode is movement vigor.

11) The results of the paper are all centered on decoding grip-force. The authors suggest that this decoding could be used as a control signal for adaptive deep brain stimulation in patients with Parkinson's disease, increasing stimulation when higher movement "vigor" is predicted (summarized from lines 346-348). It seems that this would entail a fast algorithmic approach to adapting stimulation, making adjustments in response to frequently changing movement states. A comparison to different algorithmic approaches (for example, tracking slower fluctuations in medication state) would be helpful to have in the Discussion.

We appreciate this comment and try to address it without delving too deep into speculation. Before that, we would like to state that we are convinced that adaptive neuromodulation will improve clinical outcome in DBS patients and that brain signal decoding will extend the clinical utility of adaptive DBS, and that ECoG signals outperform LFP from DBS electrodes (e.g. see Merk et al., 2022 https://doi.org/10.1016/j.expneurol.2022.113993 on the topic or Neumann & Rodriguez-Oroz https://doi.org/10.1002/mds.28567 for a short movement disorders editorial). Therefore, we are confident that the present paper is a relevant albeit indirect contribution towards the implementation of intelligent adaptive DBS beyond the present use case of PD and grip-force. In the discussion we tried to highlight the general relevance of the paper, while also trying to sketch the potential specific utility of our findings in PD. Here, it is our personal belief that stimulation proportional to movement vigor may have merit in the treatment of Parkinson’s disease as it could mimic the invigorating effects of phasic dopamine signals that are lost to neurodegeneration (see reply to comment 2). However, we want to be transparent that there is currently no clinical evidence for this use-case. Now the reviewers are right that indeed this use of vigor decoding for adaptive DBS would be a fast algorithmic adaptation to concurrent behaviour. This could be combined with a slower adaptation algorithm, e.g. based on subthalamic β activity or corticosubthalamic coherence, to rebalance tonic fluctuations of dopamine with medication cycles and or additional decoding applications, similar to the overlapping temporal scales for closedloop DBS pictured in Tinkhauser & Moraud, 2021 https://doi.org/10.3389/fnins.2021.734186. Our vision is that in the future, multiple decoders will decode concurrent symptoms, sideeffects and behaviours that are relevant for DBS adaptation in PD, e.g. bradykinesia, gait freezing, voluntary movements, dyskinesia, tremor, speech, dysarthria, dystonia and more. This vision is discussed in a previous review paper in Neumann et al., 2019 https://doi.org/10.1007/s13311-018-00705-0.

The Discussion section now reads:

“Notably, the proposed adaptive stimulation would require a fast algorithmic adaptation of stimulation to ongoing behavior. This could be combined with additional slower adaptations in response to medication or sleep cycles. Specifically for PD, β activity based adaptive stimulation can be well suited to track the patient’s overall symptom state (Tinkhauser and Moraud, 2021) while more rapid stimulation adaptations based on vigor can follow fast kinematic changes. The utility of vigor-based stimulation and the combination of this approach with additional slower adaptation algorithms, require further proof-of-concept studies before the clinical utility can be foreseen.”

12) A Go/No-Go task is also used in the paper. This is potentially problematic, given that there may be modulation in the neural recordings associated with response inhibition. While the decoders are trained only to predict actual force production, it is possible that inhibitory responses could influence the decoder readout and impact the performance metrics on which every analysis of the paper hinges. It is also plausible that such inhibitory responses would asymmetrically affect the subthalamic decoders and the cortical decoders. Comparison of those two signal locations is the main topic of the paper, so this is a critical aspect to consider and discuss.

We acknowledge the general relevance of this comment, but are convinced that the implications for the validity of our study remain limited. The absence of specific differential dynamics of EEG vs. STN recodings were addressed in a Go/No-Go task in Klostermann et al., 2007 https://doi.org/10.1111/j.1460-9568.2007.05417.x. The only specific difference reported for cortex vs. STN was an increased β rebound synchronization after movement termination, which is generally also observed in cortex. More recently and based on the same dataset and the same task the interplay of STN and cortex were evaluated, indicating an interplay of both structures during motor preparation and performance (Alhourani et al., 2020 https://doi.org/10.1093/cercor/bhz264; Fischer et al., 2020 https://doi.org/10.7554/*eLife*.51956). Finally, if decoding should work in naturalistic settings, the decoders would be required to be even more robust to distraction and additional information coding. Thus, we believe this could be interpreted to support the utility of our decoding framework during complex human behavior. Moreover, the preliminary analysis on new data outside of a Go/No-Go task shown in comment 10 can further serve as validation that the task is not required for the demonstration of higher decoding performances from ECoG recordings when compared to subthalamic LFP. Nevertheless, we now note this point as a limitation in our Discussion section, which now reads:

” Our analysis is retrospective in nature and the data were obtained in context of a Go/NoGo task, which may have implications on the generalizability of the findings in the application during naturalistic behavior.”

13) The results identify a stark difference in the way model "complexity" impacts decoding in the STN and cortex, yet this concept is not well-developed or discussed. What does this say about the nature of the information content in each area such that allowing for non-linear relationships benefits decoders based on one area, but harms decoders based on the other? Please discuss further.

In the present analysis we found that XGBOOST outperforms linear methods, such as Wiener Filters or linear regression of spatial features obtained with Source Power Comodulation (SPoC). Since XGBOOST makes use of highly non-linear feature interactions, we show that ECoG band power features encode grip force non-linearly. In the STN however linear methods seem to outperform non-linear methods, with Wiener Filters demonstrating best performances. As mentioned in our reply to comment 4, it is difficult to state a clear difference in the underlying representation of grip-force in the respective neural population, given the difference in electrode shape and the relation to the neural architecture that is recorded with ECoG vs. STN. Therefore, we like to refrain from such speculations and limit our interpretation to the empirical finding that the more complex XGBOOST models in combination with ECoG signals outperformed decoding from Wiener Filters with STN signals. We are afraid that signal to noise ratio may pose a bias that is difficult to control for. This was acknowledged in the discussion as part of our reply to comment 3 above. We have now extended this to:

“Most importantly, the signal to noise ratio may further impact decoding accuracies differently for ECoG and LFP signals. This could in part explain why decoding from ECoG signals may benefit more from complex and non-linear model architectures.

14) The Bayesian Optimization hyperparameter search was run for a fixed number of iterations according to line 637. Were enough iterations used to reach convergence in performance? This would be a useful supplementary analysis.

We thank the reviewers for identifying this important methodological aspect. Bayesian optimization was run for 10 rounds in total. We have now performed the requested supplementary analysis as shown in Author response image 7.

**Author response image 7. sa2fig7:** 

Author response image 7 shows the normalized XGBOOST performance change with incremental Bayesian optimization rounds for ECoG and STN contacts for contra – and ipsilateral movements. Importantly, the optimal hyperparameters have been found for every contact after 5 rounds. We have now added this information to the methods section, which now reads:“Post-hoc assessment confirmed convergence in performance after a maximum of 5 rounds in all recordings.”

15) In line 648, it is stated that a single best channel was chosen for each patient and brain area (STN, cortex) in part to "account for varying numbers of available electrode contacts". There seem to be far more ECoG channels than STN channels – does this introduce a bias in the results when taking the "best" performing channel from a larger candidate pool of ECoG channels as compared to the STN? The authors should elaborate on this and make any necessary corrections for bias.

We acknowledge the potential bias arising from the difference in number of ECoG and STN-LFP contacts. After bipolar rereferencing there were 3 STN channels, while on average 10.18±11.29 left and 8.9±12 right hemispheric ECoG channels. To account for this difference, we repeated the analysis by randomly selecting n=3 ECoG channels for every subject. After selecting the best cross-validated performance contact per subject for ECoG and STN-LFP, ECoG still outperforms STN-LFP recordings.

**Author response image 8. sa2fig8:** 

We have decided to abstain from including this additional analysis because we feel that for the main manuscript a more systematic approach comparing all machine learning architectures would be adequate but clutter the manuscript with technical details. Nevertheless, our results are further supported by the preliminary analysis on the Berlin subject where 6 ECoG contacts were compared to 8 STN contacts. Here, even the weakest ECoG channel outperformed the best STN LFP channel. We have now noted the potential bias in the limitations section of the discussion, which now reads:“The comparability of ECoG and LFP recordings was further affected by the higher number of available ECoG channels, when compared to only three bipolar LFP channels. However, the large effect size of superior decoding performances with ECoG may indicate that this bias does not relevantly impact the interpretation of our findings.”

16) The significantly connected fibers are shown in Figure 7B. However, due to the small nature of this sub-figure and only 1 slice shown, it is difficult to fully understand where these fibers tend to go (appears to be a combination of M1 and SMA/PM?). It may be helpful to describe in the text of the results where these fibers are going. For instance, Figure 7C gives a great look into the visualization of the topography of decoding performance. Although that type of figure may not be possible with the fiber tract analysis, at least a text description would be helpful for the reader and allow further interpretation of those results.

We appreciate the interest of the reviewers in the network topography associated with decoding performance in figure 7. The fiber tracts depicted in panel B indeed reflect projections spanning sensory, motor and prefrontal cortical regions and include descending fibers of the corticospinal tract. However, SMA even though expected from its known contribution to motor control and relevance in EEG based brain computer interfaces, was not particularly highlighted in this network. However, SMA but not preSMA is part of the functional network shown in figure 7C. Differences may arise from the limitations of diffusion MRI methodology or from the fact that fMRI better captures polysynaptic connections.

We have now added a short description of the fiber tracts to the Results section as suggested:

“The specific fiber distributions included structural projections spanning sensory, motor and prefrontal cortex, and could significantly predict decoding performance of left out channels (ρ = 0.38, p < 0.0001; thresholded at a false discovery rate α = 0.05) and patients (ρ = 0.37, p < 0.0001) in a cross validated manner (Figure 7D).”

17) The supplementary figure provides transparency regarding the variability of observed grip performances. One remaining question was whether the observed variability in decoding performance across individuals and the observed correlation between decoding performance and impairment level may actually be related to the observed variability in grip force production. Do the more impaired individuals display greater variability across trials in grip force production compared to those with less impairment? And is this then creating a more difficult decoding problem for those individuals with more extreme variability compared to individuals who have consistent force production curves?

We thank the reviewers for this important comment. It addresses a common chicken and egg problem in the field of movement disorders neurophysiology. In brief, pathophysiologically relevant changes in brain signals are expected to produce symptoms but the observed neurophysiological correlates could be cause or consequence of motor symptoms. For example, it remains unresolved whether excessive β activity in the STN is cause, epiphenomenon or consequence of bradykinesia. The conclusion of our reply to comment 2 is that PD pathophysiology may impair movement coding capacity. If this was true, we would expect a change in motor performance, otherwise it would defy its own logic or would at least require creative argumentation along the lines of magical neuromuscular compensatory mechanisms that are not captured by our analysis. Therefore, we have realized the potential impact of this comment and have tried to further substantiate our interpretation through correlation analyses of grip-force variability. To this end, we calculated the variance of maximum grip-force across all movements for every patient. Next, we correlated the resulting grip-force variability with UPDRS-III scores and decoding performances across patients. Grip force variability showed net negative yet insignificant correlations with decoding performance (Rho = -0.49, p = 0.06) and UPDRS-III (Rho = 0.48 p = 0.07). Thus, grip-force variability was unlikely the sole factor, as the strongest correlation was indeed obtained for the direct association of decoding performance with UPDRS-III (Rho = -0.55, p = 0.04) as previously reported. Our claim that pathophysiology can impair decoding performance was further substantiated through the revision and our reply to comment 2 for which we analysed time spent in β bursts. We found that time spent in β bursts in the subthalamic nucleus was inversely correlated with decoding performances even in the motor preparation period (-1 – 0) before movement onset. Interestingly, increased movement variability was specifically associated with increased subthalamic β bursts in the motor preparation period (Rho = 0.72, p = 0.004) but not during movement (Rho = 0.37; p = 0.13). Notably, subthalamic β bursts for both the motor preparation period and the movement periods were also correlated with UPDRS-III (Rho = 0.63/0.56, p = 0.02/0.05). Finally, the strongest correlations of all were found for time spent in β bursts in the subthalamic nucleus with ECoG based grip-force decoding performance during motor preparation (Rho = -0.76, p = 0.004). We conclude that grip-force variability is non-significantly but positively associated with symptom severity but does not solely explain deterioration of decoding performance with PD pathophysiology. Instead, excessive synchronization of the subthalamic nucleus in low-β bursts appears to be the strongest negative predictor for grip-force decoding performance in this study. We have now added a short comment on movement variability to the methods and results of the manuscript:

Methods

“To investigate the variability of grip-force as a potential bias for decoding performance, we calculated the variance of peak force across movement repetitions.”

Results

“To investigate potential sources of bias from patient specific information on grip-force decoding performance, we performed Spearman’s correlations with the grand average from all contra -and ipsilateral decoding performances. Averaging was necessary to obtain one value per patient. Age (ρ = -0.16, p = 0.32), disease duration in years (ρ = 0.31, p = 0.17) and number of movements (ρ = -0.41, p = 0.11) and movement variability (Rho = -0.49, p = 0.06) did not reveal significant correlations.”